# PULSE: Projection-based Unlearning via Linear Speedy Entropy Maximization

## Abstract

Machine unlearning enables selective erasure of knowledge associated with specific data points from trained models without retraining from scratch. However, existing retain-data-free methods typically degrade retain accuracy by 13–50%, require access to retain data to preserve model utility, and incur high computational costs. Moreover, the majority of existing approximate unlearning methods are not designed for the black-box setting, where the unlearner has access only to the last few layers to classifier head and not to the feature extractor which is vendor locked. To address these limitations, we propose **PULSE** (Projection-based Unlearning via Linear Speedy Entropy Maximization), a retain-data-free unlearning method that performs *knowledge localization* in representation space. PULSE introduces a learnable projection matrix that can be trained jointly with the model (fully retain-data-free during unlearning) or attached post hoc to any pretrained network (requiring only a small subset of training data for efficient initialization). During unlearning, a forget-specific projection is optimized to maximize confidence on the forget set via entropy minimization. Subtracting a scaled copy of this matrix from the original projection induces a targeted entropy increase on forget samples while preserving global model utility through controlled geometric transformations of localized feature subspaces. Extensive experiments on CIFAR-10, CIFAR-100, CIFARSuper20, and ImageNet-1k across MobileNetV2, ResNet18/50, and ViT-B/16 demonstrate that PULSE achieves competitive forgetting performance while preserving model utility. It runs faster than strong baselines thereby, establishing PULSE as a scalable and practical paradigm for efficient, localized machine unlearning in both joint-training and black-box post-hoc settings.

## 1 Introduction

With the growing integration of deep learning models into everyday routines, these models increasingly access vast amounts of sensitive personal, private and copyrighted data, raising the risk of privacy breaches and making regulatory compliance a critical requirement. Once incorporated during training, the influence of any individual data point becomes deeply entangled within the model parameters, making simple deletion of the original records insufficient to erase its effects Arpit et al. (2017). This underscores the need for methods that enable the selective erasure of knowledge associated with specific data points from trained models when requested. In response, machine unlearning has emerged as a promising approach. This approach equips a trained model with the ability to selectively forget the influence of specific instances or entire classes, which are commonly termed the forget set, while preserving utility on the remaining data, which is known as the retain set, all without the prohibitive cost of retraining the model from scratch on the retain set alone. Legal frameworks such as the General Data Protection Regulation (GDPR) in the European Union Mantelero (2013), the California Consumer Privacy Act (CCPA) Goldman (2020) in the United States and the PIPEDA privacy legislation of the Privacy Commissioner of Canada (2018) in Canada mandate the *right to be forgotten*, making machine unlearning not just a technical challenge but also a legal necessity.

Machine unlearning methods can be broadly categorized into exact and approximate approaches. Among these, exact unlearning aims to completely remove the influence of the forget set. Retraining the entire model from scratch is considered the gold standard, while more efficient approaches like sharded, isolated, sliced,

and aggregated (SISA) Bourtoule et al. (2021) unlearning exist. However, exact unlearning is resource-intensive and impractical for large-scale models or real-time applications. As a result, focus has shifted to approximate machine unlearning, which aims to emulate the effect of exact unlearning without expensive retraining. However, these methods often provide weaker guarantees regarding the complete erasure of the forget set's influence, trading perfect accuracy for practical usability in real-world scenarios. Moreover, the majority of the existing approximate unlearning methods Chundawat et al. (2023a); Foster et al. (2024); Newatia et al. (2026); Gogineni & Nadimi (2024) rely on the availability of retain set during unlearning process to preserve the model utility. In practice, however, the retain set is frequently unavailable after training because of memory constraints or regulatory policies that prohibit long-term storage of the original data. Furthermore, even when the retain set can be accessed, its use substantially increases the computational complexity of unlearning. The cost grows linearly with the size of the retain set and quickly approaches the expense of full retraining. Although performing unlearning with the retain set is preferable for maintaining model utility, this requirement makes the task significantly harder in realistic deployment scenarios.

While most existing approximate unlearning methods were developed under the assumption of white-box setting with full access to model parameters, gradients, and training data, an equally important yet under-explored setting is black-box unlearning for image classification. Although some existing methods may be adaptable to black-box scenarios, they are not explicitly designed for this setting. In many realistic deployment scenarios, developers interact only with frozen feature representations (h = $f_\theta(x)$) produced by a vendor-locked proprietary image encoder. For example, a smartphone application may use embeddings extracted from a proprietary vision API to classify photos into personal albums, while an e-commerce platform may build a lightweight product classifier on top of embeddings generated by a closed-source backbone. In such systems, the downstream developer can train only the final classifier while the feature extractor remains inaccessible. This deployment paradigm is increasingly common because collecting and pre-/training on web-scale datasets is prohibitively expensive for most practitioners, whereas vendor-provided backbones are pretrained on massive and diverse datasets that produce highly transferable and generalizable visual representations. Consequently, downstream users often rely on frozen pretrained encoders and adapt only lightweight learnable non-linear and classifier heads for their specific tasks. Since the proprietary backbone is never trained on the data of the downstream user, unlearning is sufficient to be performed within consumer-controlled components such as the classifier head.

This black-box regime is substantially more challenging than the conventional white-box setting because existing unlearning methods typically rely on full-model gradients, intermediate activations, or direct parameter updates that are inaccessible when the feature extractor is frozen and proprietary. To the best of our knowledge, only one prior work, Black-Box Forgetting (Kuwana et al. (2024)), has explicitly studied black-box unlearning, focusing on prompt tuning for CLIP-style vision-language models. Consequently, the practically important setting of supervised image classification with frozen image backbones remains largely unexplored.

In response to these dual limitations, we introduce PULSE (Projection-based Unlearning via Linear Speedy Entropy Maximization), a novel retain-data-free unlearning framework that operates entirely in the black-box regime. Our main contributions are:

- **A practical retain-data-free and inherently black-box-compatible unlearning method.** PULSE is explicitly designed for black-box unlearning: it performs unlearning exclusively by optimizing a lightweight ($d \times d$) projection matrix placed between the frozen feature extractor and the classifier head, using only the forget set. The backbone parameters remain completely frozen during unlearning. Once the projection matrix is initialized (jointly or post-hoc), PULSE requires neither retain data nor access to the vendor-locked backbone, making it highly efficient and well suited for privacy-sensitive and resource-constrained deployment settings.

- **Comprehensive evaluation on white-box and black-box settings.** We systematically evaluate PULSE under both black-box and white-box access regimes. In the black-box setting, the image encoder is vendor-locked and frozen (e.g., frozen ImageNet-pretrained ResNet18 embeddings), and unlearning is performed solely within the consumer-controlled downstream components (projection matrix and classifier head) on CIFAR-10. In the white-box setting, we consider two practical sce-

narios: (i) models jointly trained with the projection matrix, and (ii) already-trained models where the projection matrix is introduced post-hoc.

- **Scalable evaluation across models, datasets, and unlearning scenarios.** We conduct extensive experiments across CIFAR-10/100, CIFARSuper20, and ImageNet-1k using multiple architectures, including MobileNetV2, ResNet18/50, and ViT-B/16. PULSE is evaluated under diverse unlearning settings, including single-class, multi-class, sub-class, incremental, and random-sample unlearning, while consistently demonstrating strong performance across varying data scales, model capacities, and both white-box and black-box access regimes.

- **Representation-space unlearning through confidence structure inversion.** PULSE performs localized selective erasure directly in the representation space. We provide qualitative evidence via UMAP visualizations showing that forget samples lose cluster structure and become dispersed after unlearning.

## 2 Related Work

In this section, we briefly review existing work on approximate machine unlearning, highlighting their strengths and limitations to motivate our approach.

### 2.1 Machine Unlearning

Machine unlearning originated as a data-forgetting problem in statistical query learning Cao & Yang (2015). Early work focused on convex models. Guo et al. Guo et al. (2020) proposed certified unlearning for linear models, while Bourtoule et al. Bourtoule et al. (2021) introduced the SISA framework, which shards and slices the training data so that unlearning a sample requires retraining only the affected shard from a saved checkpoint. Although SISA reduces retraining cost compared to training from scratch, it incurs substantial storage overhead and becomes expensive when many samples must be forgotten. These approaches are effective for small-scale or convex problems but are generally impractical for deep neural networks.

### 2.2 Unlearning in Deep Neural Networks

Early DNN unlearning methods include fine-tuning on the retain set A. et al. (2020), gradient ascent on the forget set (Negative Gradient) A. et al. (2020), and weight perturbation scaled by the Fisher Information Matrix (Fisher Forgetting) A. et al. (2020). Amnesiac Unlearning L. et al. (2021) subtracts logged batch updates containing forget samples, while partial amnesiac unlearning Gogineni & Nadimi (2024) stores partial parameters per batch to improve efficiency. More recent methods address computational challenges: Impair-Repair A. K. Tarun & Kankanhalli (2023) corrupts class-relevant weights with error-maximizing noise and repairs with retain data (class-level only); BadTeacher Chundawat et al. (2023a) uses dual-teacher distillation, forcing the student to mimic a randomly initialized "bad teacher" on forget samples while aligning with the original model on retain samples; and SSD Foster et al. (2024) dampens parameters based on their importance estimated via the diagonal Fisher Information Matrix of the forget set. SCRUB Kurmanji et al. (2023) updates model parameters by combining gradient ascent on forget data with knowledge distillation on retain data to achieve a balance between forgetting and utility preservation. SalUn Fan et al. (2024) identifies salient weights for the forget set using gradient information and performs targeted unlearning updates on those parameters. While these methods achieve strong unlearning performance, most remain computationally expensive and rely heavily on access to the retain set during unlearning, limiting their practicality in memory-constrained or privacy-sensitive deployments.

Black-box unlearning has received far less attention. Although some existing methods could be adapted by freezing the backbone and updating only the classifier head, they were not designed for this constrained regime. To the best of our knowledge, the only prior work that explicitly studies black-box unlearning is Black-Box Forgetting Kuwana et al. (2024) (NeurIPS 2024), which performs selective forgetting in CLIP-style vision-language models via derivative-free prompt tuning. This approach is limited to zero-shot VLM settings and still requires retain-class samples. The practically important case of black-box unlearning for standard supervised image classification with frozen image backbones remains largely unexplored.

## 3 Methodology

In this section, we introduce our notations and formalize the problem setting necessary for retain data free machine unlearning. Subsequently, we propose our PULSE unlearning methodology. We also provide a mechanistic analysis in Appendix A.4, showing that entropy minimization drives the projection matrix to concentrate along classifier-aligned spectral directions associated with forget features. This induces anisotropic sensitivity and a low effective-rank structure, enabling PULSE to selectively suppress forget-specific logits through spectral subtraction while largely preserving retain-set representations and downstream utility.

### 3.1 Notations

Let $\mathcal{D} = \{(x_i, y_i)\}_{i=1}^N$ denote the training dataset, where each input image $x_i \in \mathbf{R}^d$ has a corresponding class label $y_i \in \{1, \ldots, K\}$. Let $\phi_\theta : \mathcal{X} \to \mathbf{R}^K$ be a model parameterized by $\theta \in \mathbf{R}^m$, where $\mathcal{X} \subseteq \mathbf{R}^d$ is the input space. Specifically, $\hat{y} = \phi_\theta(x) \in \mathbf{R}^K$ represents the predicted probability across classes for input $x$, and $[\hat{y}]_k$ denotes the predicted probability that $x$ belongs to class $k$.

In the retain-data-free unlearning setting, the training dataset is partitioned into a forget set $\mathcal{D}_f$ and a retain set $\mathcal{D}_r$, such that
$$\mathcal{D} = \mathcal{D}_f \cup \mathcal{D}_r, \qquad \mathcal{D}_f \cap \mathcal{D}_r = \emptyset.$$
During unlearning, only samples from $\mathcal{D}_f$ are accessible, while $\mathcal{D}_r$ is unavailable due to privacy, storage, or deployment constraints.

The goal of machine unlearning is to transform the original model $\phi_\theta$ into an unlearned model $\phi_{\theta'}$ such that the influence of the forget set $\mathcal{D}_f$ is effectively removed, while maintaining utility on the retain distribution associated with $\mathcal{D}_r$. Ideally, the behavior of $\phi_{\theta'}$ should approximate that of a model retrained from scratch using only the retain set $\mathcal{D}_r$.

### 3.2 Problem Setting

We study machine unlearning under two practical model-access regimes that reflect real-world deployment constraints. Importantly, in all experiments and both regimes, PULSE performs unlearning exclusively by optimizing a lightweight learnable projection matrix placed between the (frozen) feature extractor and the classifier head; the backbone parameters are never updated. We first present results in the black-box setting, where the image encoder is vendor-locked and inaccessible. Only the consumer-controlled components, the projection matrix and classifier head built on top of the fixed feature representations can be modified during unlearning. We further conduct extensive evaluations in the white-box joint-training setting, where the backbone, projection matrix, and classifier are initially trained together from scratch. Even in this setting, however, unlearning is performed solely by updating the projection layer while keeping the backbone completely frozen. This design makes PULSE inherently compatible with black-box deployment scenarios while still enabling fair comparison against existing white-box baselines.

### 3.3 Overview

In image classification, neural networks typically consist of a feature extractor $f_\theta(x)$ and classifier $f_\psi(x)$. Our proposed PULSE extends this architecture by introducing a learnable projection matrix $P_L$ that operates between the feature extractor and classifier during training and unlearning phases. During training, we optimize the entire model using standard classification objectives. For unlearning, we freeze all parameters except learnable projection and minimize entropy on the forget set $D_f$, forcing the projection matrix to learn highly confident representations for samples to be forgotten. We then compute the unlearning projection $P_{UL} = \alpha P_L - (1 - \alpha) P_{forget}$, where $P_{forget}$ represents the learned confident mappings. This inversion maximizes entropy on $D_f$, effectively erasing knowledge associated with the forget set while preserving performance on retained data.

The complete methodology comprises two phases: the training phase (Section 3.4) establishes the baseline model and initializes $P_L$, while the unlearning phase (Section 3.5) performs selective knowledge erasure through projection matrix manipulation.

### 3.4 Training Phase

PULSE achieves effective unlearning by performing fine-grained control over feature transformations without disrupting the entire learned representation. Traditional unlearning methods either retrain the full model or apply coarse-grained parameter updates that can significantly degrade model utility, particularly in retain-data-free settings. By introducing a projection matrix ($P_L$) between the feature extractor and classifier, PULSE creates a controllable information bottleneck for selectively filtering forget-related feature directions while preserving retain-relevant representations. This dedicated projection pathway enables targeted feature suppression during unlearning without modifying the underlying backbone or broadly perturbing the representation space, while remaining sufficient to effectively erase forget-specific knowledge from the downstream representations.

Given an input $x$, let $z = f_\theta(x) \in \mathbb{R}^d$ be the feature embedding generated by the backbone network. PULSE introduces a projection matrix $P_L \in \mathbb{R}^{d \times d}$ that operates on $z$ before the classification head $h_\psi$:

$$\hat{y} = h_\psi(P_L z),$$

where $\hat{y} \in \mathbb{R}^K$ represents the predicted logits over $K$ classes.

During training, $P_L$, $f_\theta$, and $h_\psi$ are jointly optimized to maximize classification performance on the full training set $\mathcal{D}$, using the cross-entropy loss:

$$\mathcal{L}_{\text{CE}} = -\frac{1}{|\mathcal{D}|} \sum_{(x_i, y_i) \in \mathcal{D}} \sum_{c=1}^{K} \mathbf{1}_{[y_i = c]} \log p_c^{(i)}, \tag{1}$$

$$p^{(i)} = \text{softmax}\big(h_\psi(P_L f_\theta(x_i))\big), \tag{2}$$

where $\mathbf{1}_{[y_i = c]}$ is 1 if $y_i = c$ and 0 otherwise.

We initialize the projection matrix $P_L$ as the **identity matrix**, ensuring it initially acts as a neutral transformation that leaves the feature embeddings $z = f_\theta(x)$ unchanged. This initialization strategy allows the classification head $h_\psi$ to receive the raw embeddings at the start of training, enabling the model to learn meaningful projections gradually through optimization while stabilizing training by avoiding sudden transformations of the feature space.

This architectural design enables efficient unlearning through representation space manipulation. By introducing $P_L$, we can selectively transform how learned representations are projected before classification without retraining the entire network. This approach provides faster yet robust unlearning by manipulating only the feature space through $P_L$ while keeping both the feature extractor $f_\theta$ and classification head $h_\psi$ frozen during unlearning, preserving their learned mappings while enabling selective forgetting through geometric transformations of the features. The unlearning phase, described in Section 3.5, details how $P_L$ is adjusted to achieve selective forgetting. For additional computational efficiency, the feature dimensionality can optionally be reduced before applying the projection matrix by introducing a lightweight nonlinear transformation layer. This allows ($P_L$) to operate in a lower-dimensional subspace, reducing both memory and optimization cost during unlearning.

### 3.5 Unlearning Phase

The unlearning phase fundamentally shifts from supervised cross-entropy minimization to unsupervised confidence maximization, where the optimization objective transitions from alignment with ground truth labels to amplifying the model's current posterior beliefs. This entropy-minimized projection matrix learns to sharpen posterior distributions $p(y|x)$ without external supervision, effectively implementing a self-supervised pseudo-labeling mechanism that reinforces the model's strongest predictive tendencies for the forget set.

To unlearn $\mathcal{D}_{\text{forget}}$, we freeze both the feature extractor $f_\theta$ and classifier $h_\psi$, and optimize the forget-specific projection matrix $P_{\text{forget}}$ to minimize prediction entropy on the forget set:

$$\mathcal{L}_{\text{forget}} = -\frac{1}{|\mathcal{D}_{\text{forget}}|} \sum_{(x_i, y_i) \in \mathcal{D}_{\text{forget}}} \sum_{c=1}^{K} p_c \log p_c, \tag{3}$$

where,

$$p = \text{softmax}\big(h_\psi(P_{\text{forget}}\, f_\theta(x_i))\big). \tag{4}$$

The key intuition of our approach lies in leveraging the confidence-maximizing projection as a complementary operator whose subtraction counteracts the learned confidence structure. The final projection update rule for inference combines the original and forget-specific projections:

$$P_{\text{UL}} = \alpha P_{\text{L}} - (1 - \alpha)P_{\text{forget}}, \tag{5}$$

where $\alpha \in [0, 1]$ balances retention and forgetting. Rather than directly optimizing forgetting through objectives such as gradient ascent on the forget loss or explicit entropy maximization, PULSE leverages the model's learned feature-to-class associations to induce uncertainty in a principled manner. Such objectives are often unstable in practice, exhibiting high sensitivity to optimization hyperparameters (e.g., learning rate, optimization steps, batch ordering, and stopping criteria), which can substantially degrade retain performance. Consequently, many existing approaches require an additional retain-data fine-tuning stage to recover utility.

Instead, PULSE first learns a forget-specific projection by minimizing entropy, thereby identifying the model's most confident directions for the target data. Subtracting a scaled version of this projection then performs *confidence inversion*, selectively increasing predictive entropy only for forgotten samples while preserving the discriminative structure of retained classes. During inference, the original projection $P_L$ is replaced with $P_{\text{UL}}$, enabling targeted forgetting without unstable optimization or retain-data fine-tuning.

**Applicability to Already-Trained Models:** Although PULSE is designed as a jointly trained framework where the projection matrix is optimized alongside the backbone and classifier, the unlearning mechanism described above is fully compatible with already-trained networks. In such cases, the projection layer $P_L$ can be attached post hoc to any pretrained model, initialized as the identity transform, and trained briefly on a very small subset of the original training data. Once initialized, the same unlearning procedure, optimizing $P_{\text{forget}}$ via entropy minimization and constructing $P_{\text{UL}}$ through confidence inversion can be applied without modifying any backbone parameters. This makes PULSE broadly applicable to already deployed in real-world models, enabling efficient, repeated unlearning requests without requiring full retraining.

## 4 Experiments and Results

The goal of our experiments is twofold. First, we validate the proposed method on models trained jointly with the projection matrix (Section 4.2). Second, we evaluate its applicability to already-trained models (Section 4.3). Although the primary experiments are conducted in the conventional white-box unlearning setting, where the entire model is accessible during both training and unlearning, PULSE performs forgetting exclusively by updating the projection layer while keeping both the backbone and classifier parameters frozen. This design naturally extends to black-box deployment scenarios, where only consumer-controlled components are available for modification. However, the white-box setting is adopted for the main experiments to enable fair and direct comparison with existing machine unlearning baselines, as strong black-box baselines for image classification remain limited. To further demonstrate the practicality of PULSE, Section 4.4 presents experiments in the realistic black-box setting, where the feature extractor is vendor-locked and only the downstream consumer-controlled layers are accessible during unlearning for both class-level and sub-class-level tasks. Finally, computational efficiency is evaluated through runtime comparisons with baseline methods, and qualitative analyses of the learned representation space before and after unlearning are presented in Section 4.5. Experimental details are first introduced in Section 4.1.

### 4.1 Experimental Setup

**Benchmark datasets:** We evaluate the proposed methods across four standard benchmark datasets. CIFAR-10 and CIFAR-100 Krizhevsky (2009) are included as they are widely used in the machine unlearning literature, enabling direct comparison with existing approaches. The CIFARSuper20 dataset is constructed from CIFAR-100 by merging the 100 fine-grained classes into 20 semantically coherent superclasses, following

the procedure described in Chundawat et al. (2023a). Finally, to further demonstrate the scalability of the proposed method, we include experiments on the large-scale ImageNet-1k dataset Russakovsky et al. (2015), which contains 1,281,167 training images spanning 1,000 object classes.

**Models:** We employed four widely-used ImageNet pretrained image classification architectures: ResNet18 He et al. (2016), ResNet50, MobileNetV2 Sandler et al. (2018), and Vision Transformer (ViT-B/16) Dosovitskiy et al. (2021). ResNet18 serves as a baseline, while MobileNetV2, ResNet50, and ViT-B/16 were included to illustrate the scalability of PULSE across models of increasing complexity, with parameter counts of roughly 3M, 30M, and 90M, respectively. All models were trained using a batch size of 256 with the Adam optimizer for 10 epochs. Training was conducted on an NVIDIA H100 GPU. The value of $\alpha$ was set between [0.7,0.9] selected via random search on forget set objective.

Additionally, to evaluate unlearning performance under a vendor-locked black-box feature extractor setting, we conducted experiments using the OpenCLIP ViT-B/32 model from Hugging Face. In this setup, the CLIP vision encoder was kept entirely frozen, and only a lightweight classification head was trained, simulating realistic scenarios where access to or modification of the pretrained backbone is restricted.

**Evaluation Measures:** We evaluate PULSE and the baselines using the following metrics: **Accuracy** on the forget set ($D_f$) and the retain set ($D_r$); **Similar Class Accuracy**, which measures the collateral damage on semantically related classes (e.g., "cat" when unlearning "dog" on CIFAR-10 and "flatfish" when unlearning "aquarium fish" on CIFAR-100); **Membership Inference Attack (MIA) Success Rate**, evaluated using the standard entropy-based black-box attack protocol introduced in BadTeacher Chundawat et al. (2023a) and subsequently adopted by several recent machine unlearning methods, including SSD. This metric measures whether forgotten samples can still be distinguished as training members; and **Time**, which is the wall-clock time from an unlearning request to completion in seconds. All reported results are averaged over three independent runs. For the ImageNet-1k experiments, retraining-from-scratch and MIA evaluations are omitted due to their prohibitively high computational cost, as both require repeated large-scale training or MIA evaluation for each unlearning request.

**Baselines:** We compare our method against several established baselines. *Retrain* serves as the gold standard: the model is retrained from scratch using only the retain set $D_r$, providing an ideal upper bound on accuracy over $D_r$ and a lower bound on accuracy over the forget set $D_f$, albeit at significantly higher computational cost. *BadTeacher* is evaluated using the official implementation provided by the authors of Chundawat et al. (2023a); it leverages a combination of a "good" teacher and a "bad" teacher to perform unlearning using the forget set along with a subset of the retain set. *SSD* is implemented using the authors' code from Foster et al. (2024), where model parameters are dampened according to their relative importance measured via the Fisher Information Matrices (FIMs) of $D_r$ and $D_f$. In addition to these main baselines, we also compare our method against several zero-shot unlearning approaches namely *EMMN* Chundawat et al. (2023b), *Boundary Unlearning* Chen et al. (2023) and *Just-In-Time Unlearning* Foster et al. (2025) with results reported in the appendix.

### 4.2 Unlearning with Jointly Trained Projection Matrix

**Single and Multi-Class Unlearning:** To demonstrate the effectiveness of PULSE in full-class unlearning, including both single-class and multi-class scenarios, we conducted experiments on the CIFAR-10 and CIFAR-100 datasets using ResNet50, MobileNetV2, and ViT-B/16 models. The corresponding test-set results, including forget-set accuracy and retain-set accuracy, are summarized in Table 1. For comparison, we also implemented BadTeacher and SSD as baseline methods.

The results in Table 1 and Table 13 demonstrate that PULSE achieves strong and consistent forgetting across all datasets and architectures. On the forget test set ($D_f$), our approach reduces recognition of the target classes to nearly 0% on both CIFAR-10 and CIFAR-100. Details of the comparison with other retain-data-free methods are presented in Table 12 in the appendix, where PULSE substantially outperforms existing approaches while preserving model utility. Since PULSE is a retain-data-free method, we additionally evaluate BadTeacher both with and without retain data for a fair comparison. When BadTeacher is applied without retain data, it reduces retain-set utility by up to 30%, whereas PULSE achieves performance comparable to the retraining baseline.

Table 1: Performance of the proposed PULSE for single-class unlearning on CIFAR-100 using MobileNet-V2, ResNet-50, and Vision Transformer (ViT-B/16).

| Model | Metric | Accuracy ($D_f \downarrow$, $D_r \uparrow$) | | | | | | Similar Class Acc (%) | | |
|---|---|---|---|---|---|---|---|---|---|---|
| | | Orig. | Retrain | BadTeacher (with $D_r$) | BadTeacher (without $D_r$) | SSD | **PULSE** | BadTeacher (without $D_r$) | SSD | **PULSE** |
| MobileNetV2 | $D_r$ | 76.86 | 75.85 | **77.28** | 67.69 | 46.34 | 74.34 | 49.10 | 47.17 | **50.66** |
| | $D_f$ | 82 | 0 | 2 | 0 | 0 | **0** | | | |
| ResNet50 | $D_r$ | 74.06 | 72.57 | **74** | 53.69 | 50.22 | 71.41 | 32.00 | 33.40 | **51.84** |
| | $D_f$ | 82 | 0 | 0 | 3.0 | 2 | **0** | | | |
| ViT-B/16 | $D_r$ | 79.47 | 80.74 | **80.70** | 73.23 | 79.00 | 78.61 | 40.00 | 55.70 | **56.10** |
| | $D_f$ | 90.00 | 0 | 2 | 0 | 0 | **0** | | | |

To further examine scalability, we applied PULSE to multi-class unlearning on CIFAR-100 with ViT-B/16, varying the forget-set size from 5% to 20% of the training classes. The corresponding test-set results are summarized in Table 2. PULSE demonstrates strong forgetting effectiveness across all scales, achieving forget-set accuracies of 0.59%, 0.09%, 2.92%, and 4.75% for 5, 10, 15, and 20 forgotten classes, respectively, significantly outperforming BadTeacher (up to 27.30% residual accuracy) and SSD (0.78%–25.57% inconsistent performance). While retain accuracy naturally decreases as more classes are forgotten, PULSE maintains competitive retain-set performance (77.12%–79.11%), closely approximating the original model while demonstrating superior utility preservation compared to baseline methods.

Table 2: Performance of the proposed PULSE method for multiclass unlearning on CIFAR-100 using ViT-B/16, compared against the Retrained, BadTeacher and SSD as baselines.

| Method | 5 Classes (5%) | | | 10 Classes (10%) | | | 15 Classes (15%) | | | 20 Classes (20%) | | |
|---|---|---|---|---|---|---|---|---|---|---|---|---|
| | $D_r$ | $D_f$ | Time | $D_r$ | $D_f$ | Time | $D_r$ | $D_f$ | Time | $D_r$ | $D_f$ | Time |
| Original | 80.12 | 75.65 | - | 78.32 | 76.97 | - | 79.08 | 73.00 | - | 78.89 | 75.03 | - |
| Retrain | 80.42 | **0.00** | - | 82.55 | **0.00** | - | 82.85 | **0.00** | - | 84.14 | **0.00** | - |
| BadTeacher | **81.49** | 8.73 | 72.82 | **81.24** | 27.30 | 79.60 | **82.28** | 23.86 | 178.93 | **82.98** | 24.97 | 95.55 |
| SSD | 74.56 | 2.04 | 138.02 | 74.04 | 0.78 | 143.88 | 76.34 | 14.14 | 306.12 | 77.56 | 25.57 | 159.23 |
| **PULSE** | 79.11 | **0.59** | **14.01** | 77.12 | **0.09** | **53.65** | 77.49 | **2.92** | **51.29** | 77.95 | **4.75** | **43.44** |

**Sub Class Unlearning:** This setting is inherently challenging, since the objective is to forget a specific sub-class without disrupting visually similar sub-classes within the same superclass. For example, in CIFARSuper20, the task may involve forgetting 'baby' from the 'people' superclass while retaining 'boy', 'girl', 'man', and 'woman'. We evaluated this scenario using ResNet18, MobileNetV2, and ViT-B/16 models.

Table 3 demonstrates PULSE's effectiveness for sub-class unlearning across different architectures. PULSE successfully reduces forget set accuracy ($D_f$) to near zero while maintaining high retain set accuracy ($D_r$). Compared to BadTeacher and SSD baselines, PULSE consistently achieves lower forget accuracy with minimal impact on retained knowledge, while remaining 2–10× faster based on model. In contrast, BadTeacher leaves forget set accuracies as high as 10-26%, and SSD can reach up to 31%, highlighting the considerably poorer performance of these baselines. These results confirm that PULSE can selectively erase specific sub-class knowledge without degrading model utility.

**Incremental Unlearning.** Incremental unlearning evaluates a method's ability to handle sequential unlearning requests that arrive one at a time, reflecting practical privacy scenarios (e.g., GDPR "right to be forgotten") where deletion requests occur progressively rather than in a single step.

We conduct experiments on CIFAR-10 using ResNet-50, sequentially unlearning four classes (0, 1, 2, and 3) in a strictly retain-data-free setting, where no access to retained training data is permitted during any unlearning step. Table 4 reports performance after each request. Here, $\text{Acc}_{D_r}$ denotes accuracy on retained classes (higher is better), while $\text{Acc}_{D_f}^{\text{spec}}$ and $\text{Acc}_{D_f}^{\text{overall}}$ measure accuracy on the most recently forgotten class and on all forgotten classes cumulatively, respectively (lower is better).

Table 3: Performance of the proposed PULSE for sub-class unlearning on CIFARSuper20, compared against the Retrained, BadTeacher and SSD baselines.

| Model | Superclass | sub-class | | Accuracy ($D_f \downarrow$, $D_r \uparrow$) | | | | | MIA | | |
|---|---|---|---|---|---|---|---|---|---|---|---|
| | | | | Orig. | Retrain | BadTeacher | SSD | **PULSE** | BadTeacher | SSD | **PULSE** |
| MobileNetV2 | Veh2 | Rocket | $D_r$ | 84.59 | 84.98 | **85.18** | 84.66 | 80.14±1.4 | 0 | 0 | **0** |
| | | | $D_f$ | 87 | 3 | 26 | 31 | **0** | | | |
| | Electronics | Lamp | $D_r$ | 85.58 | 79.67 | **85.95** | 67.47 | 82.36±1.14 | 0 | 0.102 | **0.05** |
| | | | $D_f$ | 79 | 6 | 23 | 11 | **4±0.47** | | | |
| ResNet18 | Veh2 | Rocket | $D_r$ | 85.32 | 84.50 | **84.82** | 84.56 | 82.91±1.2 | 0 | 0 | **0** |
| | | | $D_f$ | 89 | 3 | 3 | 4 | **0** | | | |
| | Electronics | Lamp | $D_r$ | 84.74 | 85.37 | **84.59** | 77.47 | 81.37±1.8 | 0 | 0.032 | 0.032 |
| | | | $D_f$ | 80 | 11 | 11 | 8 | **4±2** | | | |
| ViT-B/16 | Veh2 | Rocket | $D_r$ | 85.25 | 85.99 | **85.82** | 82.87 | 80.84±3.5 | 0 | 0.020 | 0.020 |
| | | | $D_f$ | 83 | 1 | 10 | 0 | **0** | | | |
| | Electronics | Lamp | $D_r$ | 88.53 | 87.29 | **88.81** | 87.91 | 86.01±0.11 | 0 | 0.022 | **0.020** |
| | | | $D_f$ | 74 | 9 | 13 | 12 | **1.6±0.47** | | | |

Table 4: Incremental class unlearning results on CIFAR-10 with ResNet50 using PULSE.

| Step | Unlearned Classes | $\mathbf{Acc}_{D_r} \uparrow$ | $\mathbf{Acc}^{\mathbf{spec}}_{D_f} \downarrow$ | $\mathbf{Acc}^{\mathbf{overall}}_{D_f} \downarrow$ |
|---|---|---|---|---|
| Initial | None | 90.92 | – | – |
| After Request 1 | 0 | 90.58 | **0.00** | **0.00** |
| After Request 2 | 0, 1 | 87.06 | **0.00** | **0.00** |
| After Request 3 | 0, 1, 2 | 85.15 | **0.00** | **0.00** |
| After Request 4 | 0, 1, 2, 3 | 85.70 | **0.00** | **0.00** |

Across all unlearning steps, the proposed method consistently drives both $\text{Acc}^{\text{spec}}_{D_f}$ and $\text{Acc}^{\text{overall}}_{D_f}$ to **0.00%**, indicating effective removal of class-specific knowledge at each stage. At the same time, retained-class accuracy $\text{Acc}_{D_r}$ remains high, decreasing from 90.92% to 85.70% after four sequential unlearning operations—a reduction of approximately 5% despite the challenging retain-data-free constraint. Notably, the cumulative forget accuracy $\text{Acc}^{\text{overall}}_{D_f}$ remains at 0.00% throughout the sequence, indicating that previously unlearned classes do not regain accuracy during subsequent unlearning steps. This behavior reflects the localized and selective nature of PULSE, where unlearning operations suppress class-specific subspaces without reintroducing or interfering with representations corresponding to earlier forget requests. As a result, forgetting remains stable across time rather than accumulating residual knowledge. These results demonstrate the robustness of our approach under dynamic, incremental unlearning settings.

**Random Sample Unlearning:** In this setting, a subset of samples is selected uniformly at random from the entire training distribution and designated as the forget set. This scenario evaluates the robustness of unlearning methods when the forget set is not class-specific and may contain heterogeneous examples drawn from diverse regions of the data manifold. To assess the stability of our method under such conditions, we conduct a *500-sample random unlearning experiment on CIFAR-10 with ResNet50*. As shown in the Table 5, PULSE achieves the lowest forget accuracy ($\text{Acc}_{D_f}$) among the baselines and lower than original model, indicating some erasure have taken place. At the same time, its test accuracy ($\text{Acc}_{D_t}$) remains comparable to that of the retrained model in case of CIFAR-10, demonstrating that PULSE preserves utility even when removing a non-structured subset of training points. In case of ImageNet-1k having retrained model is infeasible due to the heavy computational demands. Thereby, comparing among baselines PULSE achieves lower forget set accuracy and lower model utility drop.

Table 5: Random sample unlearning results (500 samples) on CIFAR-10 using ResNet-50.

| Method | $\mathbf{Acc}_{D_f}\downarrow$ | $\mathbf{Acc}_{D_t}\uparrow$ | MIA $\downarrow$ |
|---|---|---|---|
| Trained | 99.79 | 93.91 | 0.90 |
| Retrained | 92.63 | 90.28 | 0.78 |
| BadTeacher | 95.65 | 92.16 | .78 |
| SSD | 97.41 | 90.82 | 0.80 |
| PULSE | 93.80 | 89.91 | 0.79 |

Table 6: Random sample unlearning results (500 samples) on ImageNet-1K using pretrained MobileNetV2.

| Method | $\mathbf{Acc}_{D_f}\downarrow$ | $\mathbf{Acc}_{D_t}\uparrow$ | Time Taken $\downarrow$ |
|---|---|---|---|
| Trained | 71.91 | 67.94 | 2880.00 |
| BadTeacher | 71.36 | 68.82 | 56.11 |
| SSD | 71.90 | 65.58 | 20.15 |
| PULSE | 69.78 | 64.52 | 0.09 |

### 4.3 Unlearning on Already-Trained Models

While PULSE is designed to jointly train the backbone and projection matrix from scratch, it can also be applied directly to an already-trained model without requiring full retraining. In this setting, we attach the trainable projection layer immediately before the classifier head, freeze all backbone parameters, and train only the projection matrix using a small subset of the original training data (3–5%) for a few epochs. This step is extremely lightweight: since the projection layer is a single linear module, it introduces negligible computational overhead.

An important advantage of this formulation is that projection-layer training is a *one-time initialization*. Once learned, the projection matrix can support *multiple future unlearning requests* without being retrained, aligning naturally with real-world deployment scenarios in which users may request deletion of several classes or samples over time. After initialization, the standard PULSE unlearning routine can be executed repeatedly for different forget sets.

To evaluate this capability, we first trained a ResNet-50 classifier on CIFAR-10 without a projection layer. We then attached the projection module, trained it on a small fraction of the dataset, and performed class-level unlearning. As shown in Table 7, PULSE maintains high retain accuracy while fully forgetting the target class across all data percentages, with the entire unlearning step taking only $\sim 7.6$ seconds. These results closely match those obtained in the joint-training regime, demonstrating that PULSE remains effective even when applied to pretrained networks and highlighting its practicality and flexibility.

Table 7: Unlearning performance of PULSE on an already-trained ResNet-50 on CIFAR-10 with varying amounts of data used to train the projection layer.

| Data | Train Time (s) | Before | | After | | Unlearn Time (s) |
|---|---|---|---|---|---|---|
| | | $\mathrm{Acc}_{D_f}$ | $\mathrm{Acc}_{D_r}$ | $\mathrm{Acc}_{D_f}\downarrow$ | $\mathrm{Acc}_{D_r}\uparrow$ | |
| 1% | 8.48 | 96.85 | 94.72 | 0 | 90.68 | 7.61 |
| 2% | 10.95 | 95.86 | 94.36 | 0 | 91.33 | 7.63 |
| 3% | 13.26 | 96.44 | 94.48 | 0 | 93.07 | 7.58 |
| 4% | 15.37 | 96.85 | 94.65 | 0 | 93.87 | 7.64 |
| 5% | 18.12 | 96.34 | 94.68 | 0 | **94.25** | 7.62 |

We further validated this pretrained-model setting at **ImageNet scale**, using ImageNet-1k pretrained MobileNetV2, ResNet-50, and ViT-B/16 models from `torchvision`. In this experiment, the backbone weights remain frozen and only the projection matrix is initialized *once* before unlearning, following the same strategy as above. This initialization is a *one-time preparation step* and is amortized across *multiple unlearning requests*. After the projection matrix is prepared, no further training is required, and each subsequent unlearning request is handled independently using the same/updated matrix. We focus on *class-level unlearning*, where one ImageNet class is designated as the forget set and the remaining 999 classes form the retain set.

Table 8: Performance of unlearning methods on a ImageNet-1k using MobileNetV2, ResNet18, and ViT-B/16. Reported time corresponds to *per-unlearning-request runtime*; PULSE includes a one-time projection initialization step that is amortized across all future unlearning requests.

| Model | Set | Accuracy ($D_f \downarrow, D_r \uparrow$) | | | | Time per Unlearning (minutes) | | |
|---|---|---|---|---|---|---|---|---|
| | | Orig. | BadTeacher | SSD | **PULSE** | BadTeacher | SSD | **PULSE** |
| MobileNetV2 | $D_r$ | 70.85 | 73.44 | 68.13 | **68.28** | 120+ | 29.20 | **0.18** |
| | $D_f$ | 92.00 | 90.00 | 0 | **0** | | | |
| ResNet18 | $D_r$ | 66.64 | 67.46 | 61.05 | **65.86** | 120+ | 36.71 | **0.25** |
| | $D_f$ | 82.00 | 72.00 | 0 | **0** | | | |
| ViT-B/16 | $D_r$ | 80.95 | 84.68 | **80.94** | 79.67 | 120+ | 73.20 | **0.31** |
| | $D_f$ | 94.00 | 98.00 | 0 | **0** | | | |

Table 9: Class unlearning results on CIFAR-10 with ResNet18 under white-box and black-box settings. The "Original" denotes the model before unlearning.

| Method | White-box Setting | | Black-box Setting | |
|---|---|---|---|---|
| | **Retain Acc. (%)** ↑ | **Forget Acc. (%)** ↓ | **Retain Acc. (%)** ↑ | **Forget Acc. (%)** ↓ |
| Original | 94.04 | 91.88 | 81.57 | 77.56 |
| Retrained | 92.15 | 0.00 | 83.44 | 0.00 |
| BadTeacher | 94.50 | 0.31 | 82.56 | 30.88 |
| SSD | 94.24 | 4.98 | 64.98 | 0.29 |
| PULSE (ours) | **94.39** | **0.00** | **80.02** | **0.00** |

**Full-Class Unlearning.** We benchmark PULSE against BadTeacher and SSD in the full-class unlearning setting. Across all architectures, PULSE consistently achieves near-zero forget accuracy ($D_f \to 0$) while preserving high retain accuracy, thereby exhibiting substantially lower utility degradation than SSD and improved robustness compared to BadTeacher. Notably, BadTeacher frequently *fails to forget*: its forget accuracy often increases rather than decreases, even when trained for extended durations. After repeated attempts with longer training schedules, we observed no meaningful progression toward forgetting; therefore, we terminate its evaluation early and denote its runtime as "120+" minutes.

In contrast, PULSE offers both *effective selective forgetting* and *significant computational efficiency*. As shown in Table 8, PULSE achieves over two orders of magnitude reduction in per-request unlearning time compared to SSD across all architectures, and exceeds BadTeacher by an even larger margin. Importantly, these gains are achieved while maintaining competitive retain accuracy on MobileNetV2, ResNet18, and ViT-B/16. Together, these results highlight PULSE as a scalable and practical unlearning solution for large, already deployed models under repeated deletion requests.

## 4.4 Black Box Unlearning

In many practical deployments, the feature extractor is provided by a third-party vendor as a frozen foundation model, and its internal parameters are inaccessible to downstream users. Instead of training the entire network end-to-end, the consumer interacts with the model purely through inference: an input image is passed through the frozen feature extractor to obtain a feature embedding, and only a lightweight task-specific classifier is trained on top of these embeddings. Consequently, any machine unlearning algorithm designed for this setting must operate exclusively on the consumer-controlled components while leaving the vendor-provided backbone completely unchanged.

**Class Unlearning:** To demonstrate that existing unlearning methods cannot be directly adapted to this realistic black-box setting, we conduct a controlled class-unlearning experiment on CIFAR-10 using ResNet18. We compare two training regimes:

- **White-box setting:** The entire network, including the ResNet18 backbone and the classification head, is trained end-to-end on CIFAR-10 for 10 epochs.

- **Black-box setting:** A frozen ImageNet-pretrained ResNet18 serves as the vendor-locked feature extractor. Images are first passed through the frozen backbone to obtain feature embeddings, after which only the consumer-controlled components—the non-linear transformation layer, the learnable projection matrix, and the final classifier—are trained for 10 epochs.

After obtaining the initial models, we perform class-level unlearning using BadTeacher, SSD, and PULSE with identical optimization settings and the same number of unlearning epochs. In the black-box setting, all methods are restricted to updating only the consumer-controlled layers, reflecting the practical deployment constraint.

The results demonstrate that existing unlearning methods do not transfer effectively to the strict black-box regime. BadTeacher reduces forget accuracy only to 30.88%, indicating incomplete forgetting. SSD achieves near-perfect forgetting (0.29%) but does so at the expense of substantial collateral damage, reducing retain accuracy to 64.98%, approximately 17 percentage points below the original model. In contrast, PULSE consistently achieves 0.0% forget accuracy in both white-box and black-box settings while maintaining competitive retain accuracy. These results highlight that existing approaches were primarily designed for scenarios in which the backbone can be updated, whereas PULSE is explicitly designed for the practical black-box setting where only downstream consumer-controlled components are available for modification.

Table 10: Sub-class unlearning results on CIFARSuper20 using frozen OpenCLIP ViT-B/32 under the black-box setting. Only consumer-controlled layers (projection + classifier) were updated during unlearning.

| Method | Retain Acc. (%) ↑ | Forget Acc. (%) ↓ |
|---|---|---|
| Original | 86.15 | 72.65 |
| Retrained | 85.75 | 7.03 |
| BadTeacher | 86.30 | 44.50 |
| SSD | 73.09 | 0.00 |
| SalUn | 77.26 | 16.40 |
| SCRUB | 76.87 | 7.81 |
| **PULSE (ours)** | **83.63** | **7.81** |

**Sub-Class Unlearning:** To further evaluate PULSE under a realistic black-box deployment scenario, sub-class unlearning experiments were conducted on CIFARSuper20 using OpenCLIP as the vendor-locked backbone. Compared to ResNet18, OpenCLIP represents a modern vision foundation model that is widely adopted as a fixed image embedding model in practical applications. The OpenCLIP image encoder remained frozen throughout both training and unlearning, with input images first mapped to feature embeddings. Only the consumer-controlled components are, the learnable projection matrix, and the final classifier were trained for the downstream task and subsequently updated during unlearning.

Following the same experimental protocol as the class-unlearning setting, BadTeacher, SSD, SalUN, SCRUB, and PULSE were evaluated under identical optimization settings while restricting all methods to modify only the consumer-controlled layers. This ensured a fair comparison under the strict black-box constraint, where the vendor-provided feature extractor remained inaccessible. To provide a comprehensive comparison, results from SalUn Fan et al. (2024) and SCRUB Kurmanji et al. (2023) were also included alongside BadTeacher, SSD, and PULSE.

The results in Table 10 show that PULSE remains highly effective under the more challenging black-box sub-class unlearning setting with a frozen OpenCLIP backbone. Unlike existing methods, which either fail to achieve sufficient forgetting (BadTeacher and SalUn) or incur substantial degradation in retain accuracy (SSD and SCRUB), PULSE reduces forget accuracy to levels comparable to retraining while preserving strong retain performance.

These findings reinforce the observations from the ResNet18 experiment. Most existing unlearning methods assume full access to update the entire model, including the backbone, making them less effective when only consumer-controlled components can be modified, as in realistic vendor-locked deployments. In contrast, PULSE is explicitly designed for this setting, operating solely on the learnable projection matrix and classifier while keeping the pretrained feature extractor frozen. Across both ResNet18 and OpenCLIP, PULSE consistently delivers an effective privacy–utility trade-off often closely matching retrained baseline, demonstrating performance across conventional CNNs and modern vision foundation models under practical black-box constraints.

### 4.5 Representation space analysis

We leveraged Uniform Manifold Approximation and Projection (UMAP) McInnes et al. (2020) to qualitatively illustrate how PULSE performs *selective unlearning* through geometry-driven transformations in the representation space. Specifically, the UMAP visualizations are generated from the feature embeddings extracted after the projection layer and immediately before the final classification layer. Figure 1 shows these embeddings before and after applying PULSE. Before unlearning (Figure 1a), retain (blue) and forget (red) samples form clean, well-separated clusters. The forget set exhibits tight intra-class cohesion and strong separation from the retain data, reflecting confident feature representations in the CIFAR-Super20 sub-class unlearning setting.

After applying PULSE (Figure 1b), the forget embeddings undergo a pronounced geometric disruption: their clusters dissolve, and points scatter broadly across the manifold. This aligns with our mechanistic analysis, where the projection matrix $P_{UL}$ induces a targeted transformation that disperses only the forget representations. In contrast, retain samples remain compact and preserve their class boundaries. Together, these qualitative observations align closely with the empirical results, demonstrating that PULSE achieves effective unlearning while preserving model utility.

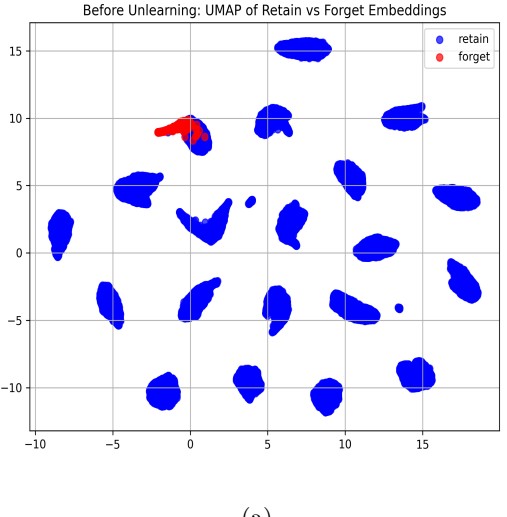
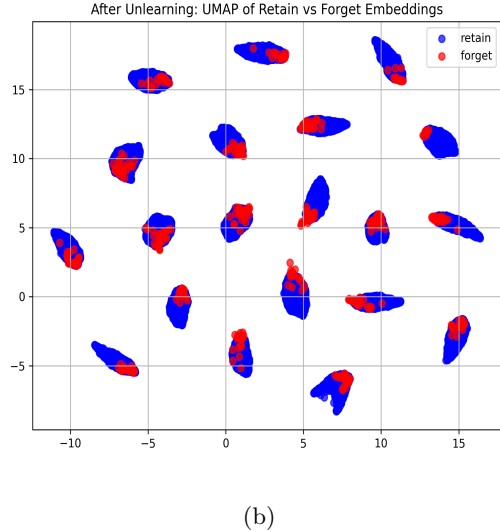

(a)         (b)

Figure 1: UMAP Embedding of sub-class Unlearning in CIFARSuper20. (a) Before Unlearning and (b) After Unlearning

## 5 Conclusion

In this work, we introduced PULSE, a machine unlearning framework based on learnable projection matrix manipulation in the representation space. Unlike most prior methods, PULSE is designed for the challenging black-box setting, where only consumer side network such as classifier head are accessible and the backbone feature extractor model remains proprietary or vendor-locked. We demonstrated its effectiveness across single-class, multi-class, sub-class, and incremental unlearning scenarios, achieving strong forgetting performance while preserving model utility across multiple architectures and datasets. PULSE operates without gradients through the backbone and supports both jointly-trained and post-hoc settings with minimal initialization data. It is also computationally efficient, providing up to $(20\times)$ faster runtime than strong baselines with low memory overhead for projection matrix. By operating exclusively on the forget set, PULSE eliminates privacy leakage concerns inherent in retain-data-dependent approaches. These advantages in empirical effectiveness, practical efficiency, and strong black-box compatibility establish PULSE as a scalable and deployment-ready solution for real-world machine unlearning, opening new directions for continual adaptation and resource-constrained applications.

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

# A  Appendix

## A.1  PULSE Algorithm

---

**Algorithm 1** PULSE: Projection-based Unlearning via Linear Speedy Entropy Maximization

---

**Input:** Trained feature extractor $f_\theta$, classifier $h_\psi$, original projection matrix $P_L$, forget set $\mathcal{D}_{\text{forget}}$, interpolation parameter $\alpha \in [0, 1]$, learning rate $\eta$, epochs $T$
**Output:** Updated projection matrix $P_{\text{UL}}$
**Training Phase:**

    1. Jointly train $f_\theta$, $h_\psi$, and $P_L$ on the full training dataset using cross-entropy loss.

    2. Freeze parameters $\theta$, $\psi$, and $P_L$ after convergence.

**Unlearning Phase:**

    1. Initialize forget-specific projection $P_{\text{forget}}$.

    2. Freeze $f_\theta$ and $h_\psi$.

    3. **for** $t = 1$ to $T$ **do**

    4.    **for** each batch $(x, y) \in \mathcal{D}_{\text{forget}}$ **do**

    5.       Compute logits: $z = h_\psi(P_{\text{forget}} \cdot f_\theta(x))$

    6.       Compute entropy loss: $\mathcal{L} = -\sum_{c=1}^{K} p_c \log p_c$ where $p = \text{softmax}(z)$

    7.       Update: $P_{\text{forget}} \leftarrow P_{\text{forget}} - \eta \nabla_{P_{\text{forget}}} \mathcal{L}$

    8.    **end for**

    9. **end for**

    10. Compute unlearned projection via confidence inversion:

$$P_{\text{UL}} = \alpha P_L - (1 - \alpha) P_{\text{forget}}$$

**Return** $(f_\theta, h_\psi, P_{\text{UL}})$ with unlearning applied.

---

## A.2  Ablation

### A.2.1  Impact of Projection Matrix Dimensions

In the proposed **PULSE** framework, the projection matrix specifies the subspace along which representations associated with the forget set are selectively attenuated. The dimensionality of this projection determines the capacity of the subspace and thus governs the trade-off between effective forgetting and preservation of retained knowledge. To analyze this effect, we perform an ablation study over the projection dimension while fixing the unlearning strength to $\alpha = 0.85$ on class level unlearning in CIFAR-10 with ResNet50. The results are presented in Table 11 for projection dimensions $\{64, 128, 256, 512, 1024\}$.

We observe that lower-dimensional projections (64 and 128) are insufficient to fully suppress representations associated with the forget classes, resulting in non-zero residual accuracy after unlearning. Increasing the projection dimension to 256 yields near-zero forget accuracy while maintaining comparable performance on retained classes. Further increases in projection dimensionality do not improve forgetting and provide marginal or inconsistent changes in retained-class accuracy.

We note that retained- and forget-class accuracies prior to unlearning vary slightly across projection dimensions. This variation arises because the projection matrix is trained jointly with the backbone model from random initialization; consequently, different projection dimensionalities induce distinct optimization trajectories due to the stochastic nature of training. Importantly, these variations do not affect the post-unlearning trends, which remain consistent across dimensions.

The interaction between projection dimensionality and the unlearning strength $\alpha$ further clarifies this behavior. For lower-dimensional projections, $\alpha = 0.85$ may be insufficient to fully suppress the forget-class sub-

Table 11: Impact of projection matrix dimensionality on retain and forget accuracy before and after unlearning (class-level unlearning on CIFAR-10 with ResNet-50, $\alpha = 0.85$). Results are averaged over 3 random seeds.

| Proj. Dim | Retain Acc. (%) ↑ | | Forget Acc. (%) ↓ | |
|---|---|---|---|---|
| | Before | After | Before | After |
| 64 | $89.81 \pm 0.32$ | $90.95 \pm 0.28$ | $94.07 \pm 0.41$ | $65.30 \pm 1.12$ |
| 128 | $93.45 \pm 0.25$ | $93.17 \pm 0.31$ | $95.95 \pm 0.38$ | $54.00 \pm 1.45$ |
| 256 | $90.77 \pm 0.39$ | $90.34 \pm 0.35$ | $93.39 \pm 0.44$ | $0.00 \pm 0.00$ |
| 512 | $91.52 \pm 0.41$ | $89.56 \pm 0.37$ | $86.27 \pm 0.52$ | $0.00 \pm 0.00$ |
| 1024 | $92.58 \pm 0.33$ | $92.04 \pm 0.29$ | $95.98 \pm 0.36$ | $0.00 \pm 0.00$ |

space, leading to residual forget accuracy. In contrast, for higher-dimensional projections ($\geq 256$), stronger unlearning steps can become overly aggressive and adversely affect retained-class representations. This suggests that higher-capacity projection spaces amplify the effect of the unlearning operation, increasing sensitivity to the choice of $\alpha$.

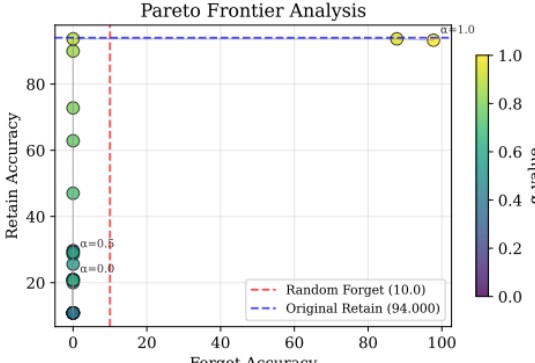

Figure 2: Pareto frontier analysis showing forget vs. retain accuracy trade-offs for different $\alpha$ values.

### A.2.2   PULSE Hyperparameter Sensitivity Analysis

To evaluate the trade-offs inherent in machine unlearning and illustrate the controllability of PULSE, we conduct a Pareto frontier analysis across different values of the hyperparameter $\alpha$ in the formulation $P_{UL} = \alpha P_L - (1 - \alpha) P_{forget}$.

Figure 2 shows the Pareto frontier of forget accuracy versus retain accuracy on CIFAR-10 with ViT-B/16. PULSE exhibits superior Pareto efficiency compared to baseline methods, demonstrating smooth and predictable control over the unlearning–retention balance as $\alpha$ varies:

**High Retention Regime ($\alpha = 0.9$–$1.0$):** Retain performance remains close to the original model with minimal forgetting—appropriate when utility preservation is prioritized.

**Balanced Regime ($\alpha = 0.7$–$0.8$):** This region yields an ideal trade-off, achieving near-zero forget accuracy ($0$–$2\%$ forget accuracy) while maintaining strong retain accuracy ($88$–$92\%$). This setting is most suitable for practical deployments.

**Aggressive Forgetting Regime ($\alpha = 0.5$–$0.6$):** Forgetting effectiveness is maximized with controlled but noticeable retain degradation, appropriate for high-privacy or security-sensitive scenarios.

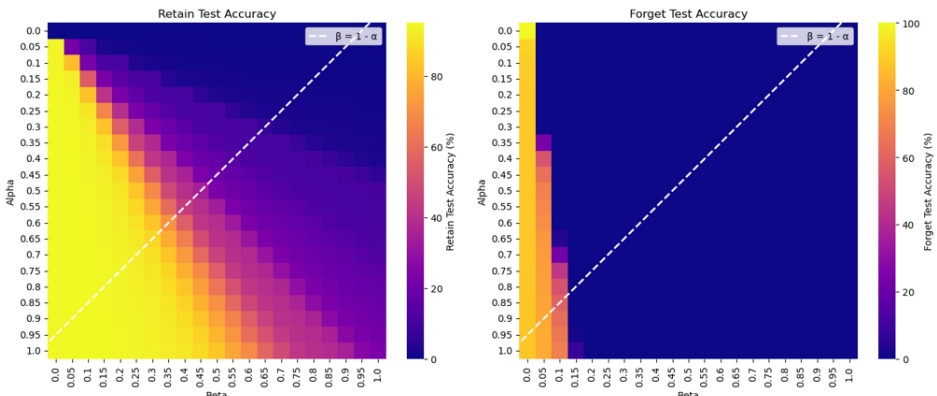

Figure 3: Exhaustive grid search over $\alpha, \beta \in [0, 1]$ for the generalized projection update $P_{\text{UL}} = \alpha P_L - \beta P_{\text{forget}}$. The left heatmap shows retain-set accuracy and the right heatmap shows forget-set accuracy. The white dashed line denotes the proposed formulation $\beta = 1 - \alpha$. The strongest privacy–utility trade-offs consistently occur on or very near this line, demonstrating that the proposed single-parameter formulation is empirically near-optimal while remaining simple and interpretable.

**Maximum Forgetting Regime ($\alpha = 0.0$–$0.4$):** Forgetting is complete at the expense of larger utility loss, suitable when forget completeness outweighs downstream performance.

Across the entire trade-off spectrum, PULSE consistently dominates baseline approaches. The Random Forget baseline (shown in red) yields inefficient outcomes, displaying both high forget accuracy ( 10%) and weaker retain performance. In contrast, every operating point of PULSE lies on or near the Pareto frontier, offering practitioners a principled and tunable mechanism for controlling unlearning aggressiveness.

This analysis highlights three key insights: (1) $\alpha$ provides an intuitive and smooth control knob for navigating the unlearning–utility trade-off; (2) PULSE maintains Pareto optimality across diverse operating requirements; and (3) the method's flexibility enables deployment across different privacy-utility preferences without algorithmic modification.

### A.2.3   Ablation on the Projection Combination Coefficients

Our proposed projection update is defined as

$$P_{\text{UL}} = \alpha P_L - (1 - \alpha)P_{\text{forget}}, \tag{6}$$

where $\alpha \in [0, 1]$ controls the trade-off between retaining the original projection and inverting the projection learned from the forget set.

Unlike a conventional convex combination, our formulation is intentionally designed as a *partial inversion*. The projection $P_{\text{forget}}$ is optimized to produce confident predictions on the forget samples. Consequently, subtracting a scaled version of $P_{\text{forget}}$ from the original projection increases the entropy of predictions on the forget set during inference, thereby promoting forgetting.

We further tie the two coefficients using a single parameter, i.e., $\beta = 1 - \alpha$, for two practical reasons. First, the resulting formulation provides an interpretable control parameter: larger values of $\alpha$ retain more of the original model, whereas smaller values increase the influence of the inverted forget projection. Second, it reduces hyperparameter tuning to a single scalar while ensuring that $\alpha = 1$ exactly recovers the original projection.

To verify whether independently optimizing the subtraction coefficient could yield improved performance, we additionally evaluated the generalized formulation

$$P_{\text{UL}} = \alpha P_L - \beta P_{\text{forget}}, \tag{7}$$

where both $\alpha, \beta \in [0, 1]$ were varied independently. We performed an exhaustive grid search over the entire parameter space and evaluated each configuration using retain-set accuracy and forget-set accuracy.

Figure 3 presents the resulting heatmaps. The white dashed line corresponds to our original single-parameter formulation ($\beta = 1 - \alpha$). The results show that the best privacy–utility trade-offs—namely, high retain accuracy together with low forget accuracy—consistently lie on or very close to this line. In particular, introducing an independent coefficient $\beta$ provides negligible improvement over the proposed formulation despite doubling the hyperparameter search space. This experiment therefore validates that the proposed single-parameter design is both computationally efficient and empirically near-optimal.

### A.3 Additional Results

### A.3.1 Comparison with Retain-Data-Free Unlearning Methods

We also examined retain-data-free unlearning methods, which attempt unlearning without access to retain data. These zero-shot approaches include: (1) **EMMN** Chundawat et al. (2023b), which generates synthetic retain data using error-minimizing noise matrices $N^{(i)}$ that minimize classification loss $L_N^{(i)}(N_r^{(i)}) = L(M(N_r^{(i)}; f), i)$ while using error-maximizing noise for forgetting; (2) **BDSH** Chen et al. (2023), which finds nearest incorrect labels via adversarial perturbations $x_f' = x_f + \epsilon \cdot \text{sign}(\nabla_{x_f} L(x_f, y, w_0))$ and shrinks decision boundaries by reassigning forget samples to these labels; and (3) **Just In Time** Foster et al. (2025), which minimizes gradients around forget points to induce boundary smoothing from an information-theoretic perspective.

However, these methods exhibit poor performance as shown in Table 12. All approaches demonstrate significant degradation in retain accuracy, with EMMN achieving only 11% (near random guessing), while failing to adequately forget target classes. These limitations motivated focusing our main evaluation on more robust retain-data-based methods. Notably, our proposed method, despite being retain-data-free, achieves performance comparable to or better than state-of-the-art retain-data-based approaches, demonstrating the effectiveness of our approach in overcoming the inherent challenges of zero-shot unlearning.

Table 12: Performance of the proposed PULSE for single-class unlearning on CIFAR-10, compared against retain-data-free methods with ResNet-50

| Method | Retain Data | $Acc_{D_r}(\%)$ | $Acc_{D_f}(\%)$ | Time (s) |
|---|---|---|---|---|
| Trained Model | ✓ | 93.69 | 94.60 | 376.61 |
| Retrained Model | ✓ | 90.10 | 0 | 342.80 |
| EMMN | ✗ | 11.2 | 0 | 50 |
| BDSH | ✗ | 80.02 | 0.29 | 103.68 |
| JiT | ✗ | 58.68 | 6.24 | 79.99 |
| PULSE (ours) | ✗ | 92.81 | 0 | 9.43 |

### A.3.2 Single Class Unlearning

### A.3.3 Details on Incremental Class Unlearning

Incremental class unlearning is a crucial capability for practical deployment, as deletion requests under privacy regulations typically arrive sequentially over time rather than in a single batch.

PULSE natively supports sequential unlearning through a compositional update mechanism on the projection layer. Let $W$ denote the original linear classification layer and $D$ the full training set. For an initial forget set $D_f^1$, the unlearned projection is computed as

$$P_{\text{UL}}^1 = \alpha P_L - (1 - \alpha) P_f^1,$$

where $P_L$ is the projection trained jointly on $D$ (as described in the main paper), $P_f^1$ is the entropy-minimized projection trained specifically on $D_f^1$, and $\alpha \in [0, 1]$ is a blending hyperparameter.

Table 13: Performance of the proposed PULSE for single-class unlearning on CIFAR-10 using MobileNet-V2, ResNet-50, and Vision Transformer (ViT-B/16).

| Model | Metric | Accuracy ($D_f \downarrow$, $D_r \uparrow$) | | | | | | Similar Class Acc (%) | | |
|---|---|---|---|---|---|---|---|---|---|---|
| | | Orig. | Retrain | BadTeacher (with $D_r$) | BadTeacher (without $D_r$) | SSD | **PULSE** | BadTeacher (without $D_r$) | SSD | **PULSE** |
| MobileNetV2 | $D_r$ | 93.75 | 91.78 | 93.73 | 90.16 | **93.20** | 90.42 | 89.60 | 91.70 | **92.10** |
| | $D_f$ | 95.04 | 0 | 13.57 | 8.92 | 3.23 | **0** | | | |
| | MIA | 0.879 | 0.4152 | 0 | 0 | 0.038 | 0.314 | | | |
| ResNet50 | $D_r$ | 93.69 | 92.48 | **93.40** | 75.26 | 92.57 | 92.18 | 72.00 | 87.20 | **94.70** |
| | $D_f$ | 94.60 | 0 | 0 | 0.30 | 0 | **0** | | | |
| | MIA | 0.880 | 0.478 | 0 | 0 | 0.128 | 0.252 | | | |
| ViT-B/16 | $D_r$ | 93.24 | 94.24 | **94.68** | 78.38 | 93.56 | 90.83 | 65.20 | **91.12** | 90.60 |
| | $D_f$ | 97.72 | 0 | 0 | 0 | 3.71 | **0** | | | |
| | MIA | 0.845 | 0.401 | 0 | 0 | 0.139 | 0.418 | | | |

For a subsequent forget set $D_f^2$, we treat the current unlearned projection $P_{\mathrm{UL}}^1$ as the new base and apply the same operation:

$$P_{\mathrm{UL}}^2 = \alpha P_{\mathrm{UL}}^1 - (1 - \alpha) P_f^2,$$

where $P_f^2$ is trained on $D_f^2$. This process can be repeated sequentially for multiple unlearning requests, yielding a final projection after $k$ requests that effectively forgets the union $D_f = \bigcup_{i=1}^k D_f^i$, without access to retained training data.

This chaining serves as an implicit merge operator: each update builds directly on the previous unlearned projection, selectively suppressing directions aligned with the new forget set while inheriting all prior edits. Because each $P_f^i$ is optimized via entropy minimization to amplify confidence specifically on its corresponding $D_f^i$, the weighted subtraction targets forget-specific subspaces with minimal interference to unrelated directions. Empirically, this preserves earlier deletions (e.g., on $D_f^1$): subsequent updates on $D_f^j$ ($j > 1$) do not reintroduce previously suppressed components unless they overlap significantly with $D_f^j$, and even then result in negligible leakage. As shown in Table 4, this mechanism empirically achieves near-zero forget accuracy on both the most recent and all prior forgotten classes at every step, with only minimal degradation on retained classes.

## A.4 Mechanistic Analysis

### A.4.1 Insights into individual projection components

Table 14 presents the performance of each projection matrix evaluated independently (by replacing the learned projection matrix in the trained model while keeping the backbone and classifier fixed). This ablation reveals the distinct roles of each component in the unlearning process.

The original projection matrix $P_L$ maintains standard classification performance with high accuracy on both forget and retain sets. Critically, when using $P_{forget}$ alone—the matrix trained to minimize entropy on the forget set—we observe increased forget accuracy (99.00%), demonstrating that this matrix amplifies confidence for the target classes as intended by the entropy minimization objective. The key insight emerges when examining $P_{UL} = \alpha P_L - (1-\alpha) P_{forget}$: the combination empirically achieves near-zero forget accuracy (0.0% forget accuracy) while preserving retain performance (92.18%).

### A.4.2 Spectral Perturbation Analysis

To determine whether PULSE induces global dampening across the representation space or a localized effect concentrated in specific directions, we analyze the spectral changes using Rayleigh quotients.

We compute the Rayleigh quotient of each eigenvector of the original covariance matrix $\Sigma$ with respect to the updated covariance matrix $\Sigma'$. This allows us to directly measure the change along each original eigen-direction after unlearning.

Table 14: Individual Projection Matrix Performance on CIFAR-10 with ResNet-50

| Projection Matrix | Forget Acc (%) | Retain Acc (%) | Interpretation |
|---|---|---|---|
| $P_L$ (Original) | 94.60 | 93.69 | Baseline trained performance |
| $P_{forget}$ (Confident) | 99.00 | 0.00 | Amplifies forget class confidence |
| $P_{UL}$ (PULSE) | 0.00 | 92.18 | Successful selective unlearning |

Figure 4: Spectral perturbation analysis showing Rayleigh quotient changes between $P_L$ and $P_{UL}$ with $\alpha = 0.8$ for CIFAR-10 class unlearning with ViT-B/16

The analysis shows that only 1.95% of the original eigen-directions experience a negative change. Moreover, the magnitude of these perturbations remains small, with a mean absolute change of 0.031 across all directions. As illustrated in Figure 4, the distribution of changes is heavily skewed toward non-negative values, with only a small number of directions showing mild decreases.

These results indicate that PULSE primarily induces localized dampening in a limited subset of directions rather than causing uniform global dampening across the representation space.

### A.4.3 Mechanistic Analysis of Entropy-Induced Spectral Concentration

This appendix provides an analysis of the entropy-based projection learning mechanism used in PULSE. Unlike earlier unlearning analyses that rely on retraining or explicit label supervision, our setting considers entropy minimization over a frozen backbone and classifier, with only a linear projection matrix optimized. We emphasize that our analysis characterizes *optimization bias* and *spectral concentration*, rather than exact rank collapse or strict feature separability.

**Setup and Notation**
Let $f_\theta : \mathcal{X} \to \mathbb{R}^d$ denote a frozen feature extractor producing representations $h = f_\theta(x)$. Classification is performed by a fixed linear head

$$z = Wh + b,$$

where $W \in \mathbb{R}^{C \times d}$ contains the class prototype vectors $w_c^\top$ as its rows and $b \in \mathbb{R}^C$ is a bias term. We introduce a learnable linear projection $P \in \mathbb{R}^{d \times d}$ applied prior to classification, yielding logits

$$z(x; P) = WPh + b, \qquad p(x; P) = \text{softmax}(z(x; P)).$$

Let $P_L$ denote the original projection matrix learned jointly with the backbone and classifier (initialized as the identity at the start of full-model training). Throughout this work, $(f_\theta, W, b)$ remain frozen and only $P$ is optimized.

**Entropy Objective and Logit-Space Gradient** For a single input $x$, the predictive entropy is

$$H(p) = -\sum_{c=1}^{C} p_c \log p_c.$$

Using the standard softmax Jacobian $\frac{\partial p_i}{\partial z_j} = p_i(\delta_{ij} - p_j)$, the gradient of entropy with respect to the logits is

$$\frac{\partial H}{\partial z_j} = -p_j\big(\log p_j + H(p)\big).$$

We collect these partial derivatives into the vector $g(p) \in \mathbb{R}^C$ defined componentwise by

$$g_j(p) = -p_j\big(\log p_j + H(p)\big).$$

When model predictions are already confident, $g(p)$ is sparse and dominated by the non-predicted (competing) classes. Consequently, entropy minimization primarily suppresses the logits of competing classes rather than further amplifying the dominant class.

**Gradient with Respect to the Projection Matrix** Applying the chain rule yields the gradient of entropy with respect to the projection:

$$\nabla_P H(x; P) = W^\top g(p(x; P))\, h^\top.$$

Thus, for each sample the gradient is a rank-one matrix formed by the outer product of a classifier-aligned direction $W^\top g(p)$ and the feature vector $h$. A gradient descent update with step size $\eta > 0$ therefore takes the form

$$P^{(t+1)} = P^{(t)} - \eta\, W^\top g(p)\, h^\top,$$

and the corresponding update to the projected feature is

$$P^{(t+1)} h = P^{(t)} h - \eta \|h\|^2 W^\top g(p).$$

Each step selectively suppresses those components of the projected feature that are aligned with competing classifier directions, thereby inducing controlled anisotropy in the representation space.

**Classifier-Aligned Spectral Concentration**
We consider the problem of minimizing the expected predictive entropy over a forget dataset $\mathcal{D}_f$:

$$\min_P \ \mathbb{E}_{x \sim \mathcal{D}_f}\big[H(p(x; P))\big].$$

**Lemma 1** (Classifier-Aligned Gradient Concentration)**.** *Assume that predictions on $\mathcal{D}_f$ are sufficiently confident and dominated by a (possibly small) subset of classes $\mathcal{K} \subseteq \{1, \ldots, C\}$. Then*

$$\mathbb{E}_{x \sim \mathcal{D}_f}\big[W^\top g(p(x; P))\big] \in \mathrm{span}\{w_k : k \in \mathcal{K}\}.$$

*Proof.* Fix an arbitrary $x \in \mathcal{D}_f$. By the assumption that the prediction $p = p(x; P)$ is confident and dominated by $\mathcal{K}$, there exists a dominant class $k^*(x) \in \mathcal{K}$ such that

$$p_{k^*}(x) \geq 1 - \epsilon, \qquad \sum_{c \notin \mathcal{K}} p_c(x) \leq \delta$$

for arbitrarily small $\epsilon, \delta > 0$ (the precise thresholds depend on the desired confidence level). Now consider the components of $g(p)$. For any class $j \notin \mathcal{K}$, we have $p_j(x) \leq \delta$. Since

$$g_j(p) = -p_j\big(\log p_j + H(p)\big)$$

and $H(p) \leq \log C$ (bounded), while $p_j \log p_j \to 0$ as $p_j \to 0$, it follows that $|g_j(p)| \leq \delta(|\log \delta| + \log C) = O(\delta \log(1/\delta))$, which can be made arbitrarily small by choosing $\delta$ small enough. Thus, $g_j(p) \approx 0$ for all $j \notin \mathcal{K}$.

Consequently,

$$W^\top g(p) = \sum_{j=1}^{C} g_j(p) \, w_j \approx \sum_{k \in \mathcal{K}} g_k(p) \, w_k,$$

i.e., $W^\top g(p)$ lies (approximately, with error controlled by $\delta$) in $\mathrm{span}\{w_k : k \in \mathcal{K}\}$. Because this holds for every $x \in \mathcal{D}_f$ and the expectation is a convex combination of such vectors, we obtain

$$\mathbb{E}_{x \sim \mathcal{D}_f}\left[W^\top g(p(x; P))\right] \in \mathrm{span}\{w_k : k \in \mathcal{K}\}$$

exactly in the limit $\delta \to 0$ (or up to an arbitrarily small residual term when $\delta$ is finite). This completes the proof. □

Lemma 1 implies that repeated gradient updates bias $P$ toward a low-dimensional subspace defined by a handful of classifier prototypes. While this does not guarantee strict rank deficiency, it induces strong *spectral concentration*. We quantify this phenomenon via the effective rank of a linear operator $A$ with singular values $\{\sigma_i\}_{i=1}^{d}$:

$$\mathrm{erank}(A) = \exp\left(-\sum_i \tilde{\sigma}_i \log \tilde{\sigma}_i\right), \qquad \tilde{\sigma}_i = \frac{\sigma_i}{\sum_j \sigma_j}.$$

Entropy minimization on $\mathcal{D}_f$ reliably drives $\mathrm{erank}(P)$ (when restricted to forget-set features) toward a small value, consistent with empirical observations.

**Stationary Bias Rather Than Convergence**
The entropy objective is non-convex in $P$, and we do not claim global convergence guarantees. Instead, our analysis characterizes the *stationary bias* of gradient descent: repeated updates reinforce the classifier-aligned directions most emphasized by $\mathcal{D}_f$. Empirically, the dominant singular vectors of the learned projection stabilize rapidly (typically within a few epochs), which is sufficient to obtain a reliable forget-specific direction for selective unlearning.

**Inversion by Subtraction and Logit Spectrum Flattening**
Let $P$ denote the projection obtained by entropy minimization on $\mathcal{D}_f$. By Lemma 1 and the rapid stabilization of dominant singular directions, the action of $P$ on forget features exhibits strong alignment with a low-dimensional classifier-defined subspace. We therefore model

$$Ph(x) \approx \beta(x)W^\top u_f, \qquad x \in \mathcal{D}_f,$$

for some vector $u_f \in \mathbb{R}^C$ and scalar $\beta(x) > 0$.

The unlearned projection is constructed via the linear combination

$$P_{\mathrm{UL}} = \alpha P_L - (1 - \alpha)P, \qquad \alpha \in [0, 1].$$

**Proposition 1** (Logit Spectrum Flattening on the Forget Set). *Under mild incoherence assumptions on the classifier prototypes (i.e., the rows of $W$ are not extremely aligned), applying $P_{\mathrm{UL}}$ reduces the variance of logits across classes for forget samples:*

$$\mathrm{Var}_c[z_{\mathrm{UL},c}(x)] < \mathrm{Var}_c[z_{L,c}(x)], \qquad \forall x \in \mathcal{D}_f.$$

*Proof.* Let $z_L(x) = WP_Lh(x) + b$ and $z_F(x) = WPh(x) + b$ denote the original and forget-specific logits, respectively. Substituting the definition of $P_{\mathrm{UL}}$ gives

$$z_{\mathrm{UL}}(x) = \alpha z_L(x) - (1 - \alpha)z_F(x) + (2 - 2\alpha)b.$$

Expanding the unlearned logits

$$z_{\mathrm{UL}}(x) = \alpha z_L(x) - (1 - \alpha)z_F(x) + 2(1 - \alpha)b$$

and computing the class-wise variance across $c \in \{1, \ldots, C\}$ for a fixed sample $x$ yields

$$\text{Var}_c[z_{\text{UL},c}(x)] = \text{Var}_c[\alpha z_{L,c}(x) - (1-\alpha)z_{F,c}(x)] + 4(1-\alpha)^2\text{Var}_c[b_c] + 4(1-\alpha)\text{Cov}_c(\alpha z_{L,c}(x) - (1-\alpha)z_{F,c}(x), b_c).$$

In our framework, $\alpha$ is chosen close to 1 (typically $\alpha \in [0.8, 0.9]$). The bias-variance term is therefore suppressed by the quadratic factor $4(1-\alpha)^2$, which equals 0.16 at $\alpha = 0.8$ and drops to 0.04 at $\alpha = 0.9$. Moreover, in well-trained classifiers the empirical variance of the static bias vector $b$ is negligible compared to the variance of the dynamic logits produced by the feature projection ($\text{Var}_c[b_c] \ll \text{Var}_c[z_{L,c}(x)]$). By the Cauchy–Schwarz inequality the cross-covariance term is correspondingly small. Consequently, the total contribution of the bias vector is negligible, allowing us to safely approximate

$$\text{Var}_c[z_{\text{UL},c}(x)] \approx \text{Var}_c\big[\alpha z_{L,c}(x) - (1-\alpha)z_{F,c}(x)\big]$$

and focus the subsequent analysis on the feature-projection dynamics. Expanding the empirical variance over the $C$ classes for fixed $x$,

$$\text{Var}_{\text{UL}} = \alpha^2 \text{Var}_L + (1-\alpha)^2 \text{Var}_F - 2\alpha(1-\alpha)\text{Cov}(z_L, z_F),$$

where $\text{Var}_L = \text{Var}_c[z_{L,c}(x)]$, $\text{Var}_F = \text{Var}_c[z_{F,c}(x)]$, and $\text{Cov}(z_L, z_F)$ are all taken across $c = 1, \ldots, C$.

Rearrangement yields

$$\text{Var}_{\text{UL}} - \text{Var}_L = (1-\alpha)\Big[(1-\alpha)\text{Var}_F - 2\alpha\text{Cov}(z_L, z_F) - (1+\alpha)\text{Var}_L\Big].$$

Entropy minimization forces $\text{Var}_F > \text{Var}_L$ (more peaked logits), while the classifier-aligned nature of $P$ (Lemma 1) ensures strong positive correlation $\text{Cov}(z_L, z_F) > 0$. Under the mild incoherence assumption on the rows of $W$, the large negative cross term dominates, so the bracketed expression is negative for any $\alpha \in [0, 1)$ whenever $P$ meaningfully reduces entropy on $\mathcal{D}_f$. Thus,

$$\text{Var}_c[z_{\text{UL},c}(x)] < \text{Var}_c[z_{L,c}(x)]$$

holds for all $x \in \mathcal{D}_f$. $\qquad\square$

Since predictive entropy $H(p)$ is a Schur-concave function maximized when the logits are equal, a reduction in logit variance tends to increase $H(p)$, especially when the original predictions are already confident (as is the case on $\mathcal{D}_f$ after entropy minimization). This flattening therefore promotes higher predictive entropy on the forget set, achieving the desired unlearning effect.

**Selectivity via Differential Sensitivity**
Selective unlearning does not require forget and retain representations to occupy strictly disjoint subspaces. Instead, we rely on a mild and realistic structural property:

**Assumption 1** (Differential Sensitivity)**.** Let $\mathcal{U}_f$ denote the dominant low-dimensional subspace induced by $P$. Then

$$\mathbb{E}_{x \sim \mathcal{D}_f}\|\Pi_{\mathcal{U}_f}h(x)\|^2 > \mathbb{E}_{y \sim \mathcal{D}_r}\|\Pi_{\mathcal{U}_f}h(y)\|^2,$$

where $\Pi_{\mathcal{U}_f}$ is the orthogonal projector onto $\mathcal{U}_f$.

This assumption holds because $P$ is optimized exclusively on $\mathcal{D}_f$ and therefore reinforces directions that are disproportionately predictive for forget samples. Consequently, for retain samples $y \in \mathcal{D}_r$,

$$P_{\text{UL}}h(y) \approx \alpha P_L h(y)$$

(with high probability when $\alpha$ is close to 1), preserving the relative ordering of logits and hence classification accuracy.

To directly validate that unlearning with PULSE increases predictive entropy on the forget set, we evaluate the effect of the interpolation parameter $\alpha$ in a homogeneous random-sample unlearning setting on ResNet-18 (CIFAR-100), where a single class form the forget set. As shown in Table 15, decreasing $\alpha$ (i.e., subtracting

Table 15: Impact of forget set predictive entropy.

| $\alpha$ | Retain Acc (%) ↑ | Forget Entropy ↑ | Forget Acc (%) ↓ |
|---|---|---|---|
| Before (original $P_L$) | 78.04 | 0.4747 | 86.20 |
| 0.98 | 78.28 | 0.5350 | 84.60 |
| 0.95 | 78.69 | 0.6416 | 81.00 |
| 0.90 | 78.91 | 0.8689 | 72.60 |
| 0.85 | 78.96 | 1.1559 | 61.90 |

a larger portion of the forget-specific projection $P$) produces a clear monotonic *increase* in predictive entropy on the forget set, rising from 0.4747 (highly confident baseline) to 1.1559.

This behavior directly confirms the core mechanistic claim of our method: subtracting the entropy-minimizing projection $P$ induces logit-spectrum flattening on forget samples (Proposition 1), which in turn drives higher predictive entropy exactly as predicted by the variance analysis. At the same time, retain accuracy remains stable or slightly improves (78.04% → 78.96%), consistent with the differential sensitivity assumption (Assumption 1).

Taken together, the classifier-aligned spectral bias, the explicit inversion construction, and differential sensitivity provide a principled mechanism for selective unlearning that operates entirely on frozen representations. This design is particularly attractive in black-box settings where only the penultimate features $h$ are accessible.

