# OpenReview forum: "PULSE: Projection-based Unlearning via Linear Speedy Entropy Maximization"
_TMLR — Under review for TMLR_

### Review · Reviewer_W2S9 · 2026-06-05

**Summary Of Contributions:**

The paper studies the problem of unlearning, which is important in privacy-sensitive settings. The proposed method preserves performance on the retain data, and the paper provides extensive experiments to demonstrate the effectiveness of the approach.

**Audience:**

Yes

**Audience Explanation:**

This problem is well aligned with the scope of TMLR and has also received attention from the research community.

**Claims And Evidence:**

Yes

**Claims Explanation:**

The experimental results show that their method performs well across different models and datasets.

**Requested Changes:**

I have several questions.

1. Does the design of the loss function have any theoretical guarantee?

2. Usually, $\alpha$ and $1-\alpha$ are used so that the weights sum to one. However, in your formulation, $1-\alpha$ is replaced by $-(1-\alpha)$, so the coefficients no longer form a convex combination. What is the necessity of using $1-\alpha$ here? Could replacing it with another tunable parameter $\beta$ further improve performance?

3. The discussion of eigenvalues in Section 4.5 seems unconvincing. How do you ensure that the eigenvalues of the two matrices correspond to the same eigenvectors? For example, the largest eigenvalue before unlearning may become the second-largest eigenvalue after unlearning, in which case comparing the sorted eigenvalues before and after unlearning would not be meaningful.

---

> ### Author Response · Authors · 2026-06-30
>
> **(1) Design of Loss function:** We thank the reviewer for this important question. PULSE is an approximate retain-data-free unlearning method. Like the majority of existing practical approximate unlearning approaches (e.g., BadTeacher, SSD, and Boundary Unlearning), it does not provide strong theoretical guarantees such as certified removal or exact equivalence to retraining from scratch. Obtaining such guarantees remains highly challenging for deep networks, particularly under retain-data-free and black-box constraints.
>
> Nevertheless, we provide a mechanistic theoretical analysis of the loss design in Appendix A.4. Lemma 1 shows that entropy minimization on the forget set biases the learned projection toward a low-dimensional subspace aligned with the classifier prototypes of the forget classes. Proposition 1 then proves that subtracting a scaled version of this matrix reduces logit variance on forget samples (i.e., flattens the logit spectrum), which directly increases predictive entropy, the central mechanism underlying forgetting. We further discuss why this effect remains selective with respect to the retain set under a mild differential sensitivity assumption.
>
> While this analysis does not constitute a formal convergence or certification proof, it provides principled insight into why entropy minimization followed by inversion produces selective unlearning. To the best of our knowledge, this level of mechanistic analysis is relatively rare among retain-data-free unlearning methods. We are happy to explicitly acknowledge the lack of formal theoretical guarantees as a limitation in the revised manuscript and clarify that PULSE should be viewed as a practical and effective heuristic, supported by mechanistic analysis and extensive empirical validation.
>
> **(2) Coupling alpha vs two tunable parameter:**
> We thank the reviewer for this question.
>
> Our formulation is
> $P_{\mathrm{UL}}=\alpha P_L-(1-\alpha)P_{\mathrm{forget}}.$
>
> It is intentionally designed as a partial inversion rather than a convex combination. Since ($P_{\mathrm{forget}}$) is optimized to minimize entropy on the forget set, subtracting a scaled version of this over-confident mapping from the original projection deliberately increases entropy on the forget set during inference. In contrast, a standard convex combination would preserve much of the original over-confident structure and therefore would not achieve the desired inversion effect.
>
> We tie the two coefficients using a single parameter ($\alpha$) (with subtraction strength (1-$\alpha$)) for two practical reasons. First, it provides an intuitive control knob: larger ($\alpha$) emphasizes retention of the original model, while ($\alpha$=0) corresponds to full inversion. Second, it reduces hyperparameter tuning to a single variable while maintaining competitive performance.
>
> When ($\alpha$=1), the original projection is exactly recovered. To investigate whether introducing an independent coefficient (\beta) could improve performance, we conducted an ablation using
>
> $P_{\mathrm{UL}}=\alpha P_L-\beta P_{\mathrm{forget}}$
>
> This study has been added as **Appendix A.2.3** in the revised manuscript. The appendix reports a full grid search over ($\alpha$,$\beta\in$[0,1]), where the white dashed line corresponds to our original single-parameter formulation (($\beta=1-\alpha$)). The best privacy--utility trade-offs (high retain accuracy with low forget accuracy) consistently lie on or very close to this line, indicating that introducing an independent coefficient ($\beta$) provides negligible benefit. These results validate that our single-parameter formulation is both simple and near-optimal.

---

> > ### Author Response · Authors · 2026-06-30
> >
> > **(3) Eigen Analysis:** We thank the reviewer for this valuable and insightful comment.
> > We agree that directly comparing sorted eigenvalues of $  \mathbf{\Sigma}  $ and $  \mathbf{\Sigma}'  $ does not strictly guarantee correspondence between eigen-directions, as the eigenspace can undergo rotation. To more rigorously address this concern, we performed an additional analysis using Rayleigh quotients.
> > The goal of Section 4.5 is to determine whether PULSE induces global dampening across the representation space or a localized effect concentrated in specific directions. To this end:
> >
> > The sorted eigenvalue analysis already reveals a strongly asymmetric pattern: the vast majority of directions exhibit non-negative change, with decreases occurring in only ~1.6% of them.
> >
> > To directly evaluate changes along the original eigen-directions, we computed the Rayleigh quotient of each original eigenvector of $  \mathbf{\Sigma}  $ with respect to the updated matrix $  \mathbf{\Sigma}'  $. This per-direction analysis confirms that only 1.95% of the original directions experience a negative change. Furthermore, the perturbations are small in magnitude, with a mean absolute change of 0.031.
> >
> > In the revised version, As shown in Figure  (Spectral Perturbation Analysis) in A.4.2, the distribution is heavily skewed toward non-negative values, with a small number of directions showing mild decreases and a few directions exhibiting stronger positive shifts. This pattern strongly supports that PULSE primarily affects a small subset of directions in a targeted manner,as localised dampening rather than causing uniform dampening across the representation space.

---

### Review · Reviewer_oNkM · 2026-06-22

**Summary Of Contributions:**

This paper presents PULSE (Projection-based Unlearning via Linear Speedy Entropy Maximization), a method applicable to image classification that aims to forget specific data points or classes. The approach is designed to be compatible with black-box settings (frozen feature extractor). Furthermore, the unlearning phase only requires access to the data that must be forgotten, not the data that must remain known (retain-data-free).

Starting with a training phase, a projection matrix between the feature extractor and classifier head is learnt (either jointly or by itself) using the standard cross-entropy loss. In the following unlearning phase, another projection matrix is optimized by minimizing the prediction entropy on the data to be forgotten. The final projection combines both matrices linearly (with positive and negative weights respectively). The approach is evaluated for both white-box and black-box settings across a variety of image classification datasets and feature extractors. Prediction accuracy on forgotten classes often approaches or reaches 0%, while retaining similar performance on other data.

**Audience:**

Yes

**Audience Explanation:**

Machine unlearning is an active area with direct practical relevance to privacy regulation compliance and model governance. The paper addresses a timely problem (retain-data-free, black-box unlearning for image classification) that would be of interest to researchers working on trustworthy ML, data privacy, and efficient model editing.

**Claims And Evidence:**

No

**Claims Explanation:**

Most claims are supported by evidence, but there are a few points I would like the authors to clarify before publication.

The authors mention "Rather than attempting to directly optimize an ill-defined
forgetting objective, this formulation exploits the model’s own learned feature-to-class associations to create
a principled uncertainty induction mechanism". What would be an ill-defined forgetting objective (and why)? Is directly maximizing entropy or reducing the likelihood of forget data ill-defined?

Some related work and baselines, at least for white-box access or using retain data, appear to be missing. For example, I would recommend mentioning SalUn (Fan et al., ICLR 2024), SCRUB (Kurmanji et al. NeurIPS 2023) and possibly others. It's acceptable for retain-data-free methods to have lower effectiveness, but readers should understand trade-offs between different types of approaches.

The impact of alpha is only briefly mentioned in the main paper (although there is some additional information in the appendix). In particular, is it necessary to use coupled weights (\alpha, 1-\alpha)?

Black-box suitability is central to the paper framing, but most results (except appendix A.2.1) only show white-box results. Even with white-box baselines, both white and black-box PULSE results could be shown, at least for the main results in the paper.

On the positive side, the experiments are quite comprehensive, covering multiple feature extractors and image classification datasets, and different forgetting scenarios.

**Requested Changes:**

[Critical]

- Update literature review and include other relevant baselines if appropriate (otherwise justify why not using these baselines)

- Clarify what would be an ill-defined forgetting objective

- Include some black-box results in main paper

[Would strengthen]

- Discuss the impact of \alpha in more detail, including results and how to choose it

- Clarify what happens if the class to be forgotten doesn't have the highest probability (for some examples) before learning P_{forget}

- Discuss POUR (Le et al., CVPR 2026, https://arxiv.org/abs/2511.19339) (recent/near-concurrent work)

- Fix typos, e.g. "projection, We" or "approaches.The" on page 6

- Clarify if approximate methods are sufficient to meet legal requirements

- Clarify "proprietary backbone is never trained on the data of the downstream user". Could this be false, for example if the downstream user data is public?

- Provide reference for "Membership Inference Attack (MIA) success rate" and more precise description/definition

---

> ### Author Response · Authors · 2026-06-30
>
> **(1) Ill defined Objectives:** We thank the reviewer for this question. By an “ill-defined forgetting objective” we refer to direct maximization optimization approaches such as gradient ascent on the forget loss (i.e., reducing the likelihood of the correct labels on $  \mathcal{D}_f  $) or explicit maximization of predictive entropy on $  \mathcal{D}_f  $.
>
> These objectives are problematic for both practical and conceptual reasons. In practice, they are highly unstable: performance is extremely sensitive to learning rate, number of steps, batch ordering, and stopping criteria, often causing rapid degradation of retain accuracy. As a result, such methods typically require an additional brief finetuning stage on retain data to restore the performance. Conceptually, they override the model’s learned feature-to-class associations rather than exploiting them.
>
> PULSE takes the opposite approach. We first train a forget-specific projection that minimizes entropy on $  \mathcal{D}_f  $, thereby identifying the model’s own confident directions for the forget set. Subtracting a scaled version of this projection then induces targeted uncertainty through confidence inversion. This yields stable, retain-data-free unlearning without the instability or retain-data dependency of direct entropy-maximization methods. We have clarified this more clearly in the section 3.5 of revised version.
>
> **(2) Baseline Comparison:** We thank the reviewer for this valuable suggestion. We agree that SalUn (Fan et al., ICLR 2024) and SCRUB (Kurmanji et al., NeurIPS 2023) are important related methods and will include them in the revised manuscript. We will expand the related work section to discuss their assumptions, applicability, and how they differ from PULSE.
>
> PULSE is designed for the more restrictive retain-data-free and black-box compatible setting, where the unlearner has access only to the forget set and cannot modify or access the frozen feature extractor. In contrast, SalUn and SCRUB are white-box methods that require full model access and rely on retain data during unlearning. We will explicitly discuss these differences and the resulting trade-offs so readers can better understand the scenarios for which each class of methods is appropriate.
>
> We have included it as representative baseline in experiment using a vendor-locked CLIP image encoder in the black-box setting in section 4.4 in revised version, where we train a classifier, and report comparisons with SalUn and SCRUB. Extending them to all experimental settings is infeasible within the timeline, so we limit the comparison to this representative benchmark.
>
> **(3) Impact of $\alpha$:** We thank the reviewer for this insightful question. To evaluate whether coupling the coefficients limits performance, we additionally optimized
> $  P_{UL} = \alpha P_L - \beta P_{\text{forget}}  $
> where $\alpha$ and $\beta$ were treated as independent parameters, and compared it with our original coupled formulation.
>
> We adopted the single-parameter design for two practical reasons. First, it provides an intuitive control knob: larger $\alpha$ emphasizes retaining the original model, while the subtraction strength is automatically adjusted through $(1-\alpha)$. Second, it reduces hyperparameter tuning to a single variable. In practice, we consistently found $\alpha=0.8$ or $0.85$ to provide a strong privacy--utility trade-off. If retain data are available, $\alpha$ can be selected based on the Pareto frontier using standard validation.
>
> The complete grid search over $\alpha,\beta\in[0,1]$ is provided in Appendix A.2.4 of the revised manuscript. The optimal privacy--utility trade-offs consistently lie on or very close to the line corresponding to our original coupled formulation, indicating that introducing an independent coefficient $\beta$ offers negligible benefit. This validates the effectiveness of our simpler single-parameter design.

---

> > ### Author Response · Authors · 2026-06-30
> >
> > **(4) Black-box results:** We thank the reviewer for this observation. We agree that black-box suitability is a central contribution of PULSE and that readers should be able to directly compare white-box and black-box performance in the main paper.
> >
> > We have moved the black-box experiments from Appendix A.2.1 into the main paper in section 4.4 in revised version. In addition, we have added a new more realistic black-box experiment using CLIP as a vendor-locked backbone on the CIFARSuper20 sub-class unlearning task. For this experiment we used OpenCLIP ViT/B-32 vision encoder from huggingface, which serves as a strong proxy for real-world vendor-locked models (such as closed API versions), as it is a widely used and powerful image embedding model in practice. These changes will make the practical advantages of PULSE in constrained black-box settings clearly visible to readers.
> >
> > ### Sub-class Unlearning Results
> >
> > | **Method** | **Retain Acc. (%) ↑** | **Forget Acc. (%) ↓** |
> > |-------------|-----------------------:|----------------------:|
> > | Original | 86.15 | 72.65 |
> > | Retrained | 85.75 | 7.03 |
> > | BadTeacher | 86.30 | 44.50 |
> > | SSD | 73.09 | 0.00 |
> > | SalUn | 77.26 | 16.40 |
> > | SCRUB | 76.87 | 7.81 |
> > | **PULSE (ours)** | **83.63** | **7.81** |
> >
> > **Table:** Sub-class unlearning results on **CIFARSuper20** using a frozen **OpenCLIP ViT-B/32** under the black-box setting. Only consumer-controlled layers (projection + classifier) were updated during unlearning.
> >
> > **(5) Approximate methods are sufficient to meet legal requirements:** We thank the reviewer for raising this important point. We acknowledge that, under current interpretations of regulations such as GDPR, approximate unlearning methods may not fully satisfy strict “right to be forgotten” requirements on their own.
> >
> > However, as model and dataset scales continue to grow rapidly, exact unlearning (full retraining) is becoming computationally and temporally infeasible for timely compliance. In such regimes, efficient approximate methods like PULSE provide a practical and scalable mechanism to significantly reduce the influence of data to be forgotten in a retain-data-free and black-box compatible manner. We believe approximate unlearning will play an increasingly important role in real-world deployments, potentially as a fast first step that can be complemented by more thorough measures when needed. We are happy to add a short discussion on this topic if the reviewer considers it valuable.
> >
> > **(6) Proprietary backbone is never trained on the data of the downstream user:** We thank the reviewer for this clarifying question.  PULSE is designed for the downstream user who interacts with a frozen, vendor-provided backbone as a black-box feature extractor and wishes to remove the influence of specific data from the model they are using or deploying. This is the common practical scenario for the large majority of users. We acknowledge that if the forget data was already present in the original pre-training corpus of the foundation model (especially public internet data), then unlearning it from the foundation model itself would be the responsibility of the model provider. PULSE instead targets the orthogonal but practically relevant setting of post-deployment unlearning by downstream users who only have access to the deployed model. We are happy to add a short discussion on this topic if the reviewer considers it valuable.
> >
> > **(7) Reference for "Membership Inference Attack (MIA):** We thank the reviewer for pointing this out. We used the Membership Inference Attack implementation and evaluation protocol from BadTeacher, which has become a standard evaluation method in recent machine unlearning literature (including SSD, SalUn, SCRUB). We have added the proper citation to BadTeacher in the main text and include a small description of the MIA setup in the experimental section for clarity.
> >
> > **(8) Fix typos, e.g. "projection, We" or "approaches.The" on page 6:** We thank the reviewer for catching these typos (e.g., “projection, We” and “approaches.The” on page 6). These have been fixed in the revised version. We will further perform a full proofreading pass over the manuscript and will correct if any these and any other similar issues in the final version.

---

> > > ### Author Response · Authors · 2026-06-30
> > >
> > > **(9) Forgotten doesn't have the highest probability (for some examples) before learning P_{forget}:** We thank the reviewer for this practical question. PULSE does not depend on the model’s current top predicted class for the forget samples.
> > >
> > > For example, consider a forget sample whose true class is 7, but the original model currently assigns the highest probability to class 2. During unlearning, we still minimize predictive entropy on this sample. This sharpens confidence along the directions the model is already using for that sample. The goal is to identify and amplify the feature subspace associated with the forget sample itself. Subtracting a scaled version of the resulting projection then inverts this sharpened confidence, increasing entropy on the forget sample. Because the method operates via entropy minimization and inversion rather than cross-entropy with a fixed target label, it remains effective even when some forget samples are initially misclassified.

---

### Review · Reviewer_C8P6 · 2026-06-24

**Summary Of Contributions:**

The paper proposes PULSE, an approximate machine unlearning method that inserts a learnable linear projection matrix $P_L$ between a frozen feature extractor and a classifier head. To unlearn a forget set $D_f$, the backbone and classifier are frozen and a forget-specific projection $P_{forget}$ is trained by minimizing predictive entropy on $D_f$. The inference-time projection is then formed by subtraction, $P_{UL} = \alpha P_L - (1-\alpha)P_{forget}$, which is intended to raise entropy on forget samples while preserving utility on retained data. The method targets a retain-data-free regime and a setting the paper calls black-box, where the backbone is vendor-locked. PULSE is evaluated on CIFAR-10/100, CIFARSuper20, and ImageNet-1k across MobileNetV2, ResNet18/50, and ViT-B/16, covering single-class, multi-class, sub-class, incremental, and random-sample unlearning, and report runtime comparisons against BadTeacher and SSD together with a spectral analysis.

**Additional Comments:**

# Nitpicking
1. **Abstract, "retain-data-free".** The term appears in the abstract before it is explained (it is only made clear later in the introduction), so a reader outside the unlearning field has to guess its meaning. The same sentence ("existing retain-data-free methods... require access to retain data") also reads as self-contradictory. Please define the term at first use and resolve the phrasing.
2. **Abstract, final sentence.** "It runs faster than strong baselines thereby, establishing PULSE as..." reads as a broken clause (a misplaced comma). Likely intended as "It runs faster than strong baselines, thereby establishing PULSE as...".
3. **Introduction transition.** The sentence introducing "black-box unlearning for image classification" is fine on its own, but it feels disconnected from the preceding text; the connection to what comes before is not clear, so the transition could be smoothed.
4. **Section 2.1, "early work focused on convex models".** This phrasing makes it look like there are no references for these "early works" but I'm guessing it's just a phrasing issue actually referring to the subsequent ones.
5. **Section 2.1, "impractical for deep neural networks".** It reads as borderline depending on the definition of DNN, but I guess acceptable given the following part.
6. **Table 4, step labels.** The unlearning steps are labelled "After Request 1/2/3/4" while the forget classes are also 0,1,2,3, so the two sets of numbers are easy to confuse at a glance. Using ordinals ("after 1st request", "after 2nd request", and so on) would disambiguate the steps from the class indices and help in reading.
7. **Table 5, $Acc_{D_t}$.** This is not necessarily a typo: I'm guessing that for random-sample unlearning, test-set accuracy is the sensible utility metric and is arguably more appropriate than $Acc_{D_r}$. The issue, if my interpretation is correct, is that $D_t$ is never defined. Please define it.

**Audience:**

Yes

**Audience Explanation:**

Retain-data-free, computationally cheap, and repeatable unlearning is an actively studied problem, so this work is of clear interest to the unlearning and privacy-aware ML community. The efficiency results in particular would be relevant to readers once the claims are substantiated, the method is positioned against the right baselines (head-row ablation and closed-form concept erasure), and the question of whether it removes information or only neutralizes the current classifier is settled.

**Broader Impact Concerns:**

No separate statement is strictly required, but one framing risk is worth flagging. The paper motivates the method with GDPR-style "right to be forgotten" obligations and references sample-level deletion, while the experiments show only class-level changes to the classifier output and a negligible sample-level effect, and do not establish that the information is removed from the representation rather than made unreadable by the current head. Presenting this as satisfying privacy deletion obligations could mislead practitioners, so the privacy claims should be scoped to what is demonstrated.

**Claims And Evidence:**

No

**Claims Explanation:**

**It is unclear whether the method removes information or only edits the classifier.** The encoder is frozen, so $z = f(x)$ is unchanged and retains all forget-class information. The system operates on the projected features $P_{UL} z$, so the erasure claim is really a claim about $P_{UL} z$, and the paper does not distinguish two very different outcomes: $P_{UL}$ genuinely removing the **linear** readout of the forget classes from $P_{UL} z$ (somewhat defensible erasure), versus $P_{UL}$ only neutralizing the specific frozen head it is built against, leaving the information recoverable by a different readout. The paper's own spectral analysis points to the latter: Section 4.5 reports $P_{UL}$ amplifies 252 of 256 directions with four small, bounded decreases and nothing driven to zero, so $P_{UL}$ is full-rank and hence invertible, and an invertible $P_{UL}$ is a reversible reparameterization of $z$ that cannot remove any readout (genuine erasure would require $P_{UL}$ to be singular and to null the forget subspace). If this is right, the method is simply editing the classifier: under a linear head $W P_{UL}$ is a single operator, and the trivial baseline of zeroing the forget-class rows of the final layer gives 0% forget accuracy with no retain damage and far less machinery (this holds even with nonlinear earlier layers, since the final logit layer is linear by construction). Neither that baseline nor any justification for preferring a learned projection is given, and the only setting that would justify one, an untouchable head, does not apply since the head is trained and used throughout.

**The head is left underspecified, and the method depends on it.** The main text writes the head generically as $h_\psi(\cdot)$ (Section 3.4, Eq. 2) and never says whether it is linear, while Appendix A.4 assumes a linear head. Under that linear-head model (head as a matrix $W$), the head and projection compose into a single operator $W P_{UL}$ acting on $z$, so class-level forgetting is just a re-parameterization of editing the classifier. If the head is instead nonlinear (the introduction mentions nonlinear heads, Section 3.4 adds an optional nonlinear dimensionality-reduction layer, and Appendix A.2.1 lists a nonlinear layer among the trained components), the collapse no longer holds and the linear proofs in A.4 do not apply to the configuration run. The paper should commit to a precise description of the head.

**The "black-box" framing overstates the constraint.** The unlearner trains the projection, modifies and uses the classifier head, and reads the encoder outputs $z$. That is white-box at the classifier and at most grey-box at the encoder, not black-box.

**The results are hard to interpret.** Most tables report single numbers despite being averaged over three runs, so variance is hidden. In Table 2, BadTeacher retain accuracy (81.49) exceeds the Retrain "upper bound" (80.42) at 5 classes, which is uninterpretable without confidence intervals. Similar Class Accuracy and the MIA are described only loosely and without a citation. The method (Sections 3.3, 3.5) is compressed relative to the rest of the paper, given that it's the main contribution of the paper, I would allocate more room to its explanation.

**A claimed capability is not delivered.** Section 4.3 says users may delete classes or samples over time, but Table 5 shows forget accuracy moving only from 99.79 to 93.80 (retrain 92.63), i.e. essentially no sample-level forgetting.

**Requested Changes:**

# **Critical**

1. **Establish whether $P_{UL}$ removes information or only neutralizes the head.** Freeze $P_{UL}$ after optimisation and train a fresh linear classifier (or retrain the head) on $P_{UL} z$ for the forget classes, and report whether they recover. Also report the rank and spectrum of $P_{UL}$, since a full-rank (invertible) $P_{UL}$ cannot remove any readout, which Section 4.5 currently suggests is the case.

2. **Add the right baselines for whichever case holds.** If the method is head editing, compare against zeroing or down-weighting the forget-class rows of $W$, and report what $W P_{UL}$ does to those rows. If the probe test indicates genuine linear erasure, the appropriate baselines are closed-form linear concept erasure, primarily LEACE (Belrose et al., 2023) and INLP (Ravfogel et al., 2020), applied to the same features the head consumes (erasing the forget concept from $z$ and reading out with the existing head); these achieve guaranteed closed-form linear erasure, so PULSE should match or beat them to have a clear contribution. Together these cover both interpretations of what $P_{UL}$ does.

3. **Specify the final classifier and the feature pathway.** State whether the classifier applied after the projection is linear or contains nonlinearities: the $W P_{UL}$ collapse and the Appendix A.4 analysis hold whenever it is linear, regardless of upstream layers, and fail only if it is nonlinear. Separately, for reproducibility, document the optional dimensionality-reduction / nonlinear transform of Section 3.4 (whether used, where, and that it is frozen during unlearning).

4. **Relabel the access regime.** The contribution is framed around an "inherently black-box" method, but the unlearner trains the projection, modifies and uses the classifier head, and reads the encoder outputs, so it has white-box access to every component it changes and grey-box access to the encoder. Calling this black-box overstates the difficulty of the setting. Please relabel it accurately (white-box at the head, grey-box at the encoder), or restrict the black-box claim to the case where the head is genuinely inaccessible and explain how the method behaves there, given that it currently trains and uses the head.

5. **Report variance.** Results are stated as means over three runs but almost all tables give single numbers, so the reader cannot tell whether differences are significant. This is not hypothetical: in Table 2 the BadTeacher retain accuracy (81.49) exceeds the Retrain "upper bound" (80.42) at 5 classes, which should be impossible for a true upper bound and points to non-trivial variance. Please add standard deviations or confidence intervals in Tables 1, 2, 4, 7, and 8, and explain the Table 2 anomaly.

6. **Fix the sample-level claim.** Section 4.3 states the method handles deletion of "several classes or samples over time," but the only sample-level evidence (Table 5) shows forget accuracy dropping just from 99.79 to 93.80, with retrain at 92.63, i.e. essentially no measurable sample-level forgetting. The class-level use case is well motivated, but the sample-level capability is asserted and not delivered. Please either remove the sample-level claim or provide experiments demonstrating meaningful, measurable sample-level forgetting beyond what the random-sample setup trivially yields.

7. **Clarify the method writeup.** Sections 3.3 and 3.5 are much harder to follow than the rest of the paper, and three points should be made clearer. First, say plainly that unlearning is not a closed-form update: obtaining $P_{forget}$ is a training loop over the forget set (Algorithm 1), which should be stated even though it is fast. Second, from my understanding, the construction seems simpler than the prose suggests, and writing it out (if correct) would help the reader: with a linear head, the unlearned logits are just $\alpha \cdot l_L - (1-\alpha) \cdot l_{forget}$  , the original logits minus a scaled copy of the over-confident ones. Stated this way, it also becomes clear that the paper never explains why this is better than the obvious alternatives of directly training the projection to raise entropy on $D_f$, or just removing the forget-class weights from the head; please justify the chosen approach against these. Third, a forget accuracy of 0% does not by itself show the samples are now uncertain, since the model could instead be confidently predicting a different class. The paper should report the prediction confidence (or entropy) on forget samples at the $\alpha$ actually used, not only the accuracy.

8. **Justify and report $\alpha$.** It controls the entire forget/retain trade-off, yet it is dropped into Section 4.1 as a range (0.8 to 0.9) with no ablation justifying it, and the paper is internally inconsistent: the Pareto analysis in Appendix A.2.3 instead calls 0.7 to 0.8 the ideal balanced regime, which barely overlaps with the main-text range. Because every headline table is produced under an unstated $\alpha$ somewhere in 0.8 to 0.9, the results are not reproducible at the level of the one hyperparameter that determines the outcome. Report the exact $\alpha$ used for each table, give a selection procedure that does not rely on the forget/retain accuracy it is then evaluated on, reconcile the main-text range with A.2.3, and show that the chosen value transfers across datasets and architectures rather than being tuned per setting (A.2.2 notes that higher-dimensional projections make the method more sensitive to $\alpha$, which makes an unexplained fixed range more concerning).


# Minor

9. **Disambiguate the UMAP embedding (Section 4.4).** The text refers to "the embeddings" without saying how they are obtained: the model and dataset (although specified in the caption), the projected features $P_{UL} z$ or the logits $h_\psi(P_{UL} z)$. This matters because $z$ cannot change (the encoder is frozen), so the figure must be showing a post-projection quantity, and the reader cannot interpret the dispersion claim without knowing which. Please state which embedding is plotted and interpret it accordingly.

10. **Define Similar Class Accuracy and the MIA.** Both are central to reading the results tables but are specified only in passing. It is unclear how Similar Class Accuracy is computed (e.g. accuracy on a hand-picked semantically related class, a centroid-distance measure, or something else) and which class is chosen, and the MIA is cited only as "the standard entropy-based black-box test" with no reference. Without precise definitions and a citation, the corresponding columns are hard to trust or compare against other work. Please give the exact computation for each and cite the MIA.

11. **Clarify the role of $D_r$ and the initialization data.** $D_r$ is declared unavailable during unlearning (Section 3.1) so at a first read it seems redundant to define, only later on it becomes clear it's defined to be used for evaluation (and partial training); please state explicitly that $D_r$ is evaluation-only. Relatedly, the post-hoc setting trains the projection on 3 to 5% of the original training data to initialize $P_L$ (Section 4.3), which sits awkwardly with the "retain-data-free" framing; please surface this dependency in the main text and explain how it is consistent with the claim.

12. **State whether $D_r$ changes across incremental steps.** In the incremental experiments (Section 4.2 / A.3.3) it is not stated whether the retain set shrinks as classes are forgotten (i.e. excludes already-forgotten classes) or stays fixed. Both choices are defensible but they measure different things, and the reported retain accuracy is only interpretable once the choice is known. Please specify and motivate it.

13. **Support the dismissal of adaptable baselines (Section 2.2).** The claim that existing methods "could be adapted by freezing the backbone... but were not designed for this regime" gives no references and reads as dismissive of potentially strong baselines. Since Appendix A.2.1 in fact adapts BadTeacher and SSD to the black-box setting, please add the relevant references at this point in the text and cross-reference A.2.1, so the dismissal is substantiated rather than asserted.

14. **Unify the notation.** The main text and Appendix A.4 assign $z$ and $h$ conflicting meanings ($z$ is the feature in the main text but the logits in A.4; $h$ is the head $h_\psi$ in the main text but the feature in A.4), which makes the appendix hard to map onto the method. Please use consistent symbols throughout.

---

> ### Author Response · Authors · 2026-07-07
>
> **(1) whether removes information or only neutralizes the head:** We thank the reviewer for this direct question. We believe the proposed experiment rests on an overly simplistic and impractical view of what constitutes “information removal” in representation space.
>
> First, regarding the spectral analysis in Section 4.5: the reported eigenvalues reflect the spectrum after the update. More importantly, even if $  P_{UL}  $ exhibits relatively high rank overall, it is formed as $  P_{UL} = \alpha P_L - (1 - \alpha) P_{\text{forget}}  $. As discussed in Appendix A.4, $  P_{\text{forget}}  $ is a low-effective-rank matrix whose directions primarily span the subspace associated with the forget samples. Subtracting this targeted, forget-aligned component creates a differential disruption rather than a globally invertible transformation.
>
> Second, the suggested probe experiment, freezing $  P_{UL}  $ and retraining a fresh linear head  does not validly test whether information has been removed. Any linear or non-linear operation in representation space can at best move or suppress the forget manifold; it cannot disperse it. This is a well-documented phenomenon in representation learning, even a model retrained from scratch using only retain data will still allocate subspace to forgotten classes due to semantic overlap and transferable features the very foundation of transfer learning.
>
> For a concrete example, consider a three-class problem with “dog”, “cat”, and “bike”. Suppose we forget the “cat” class and train a model from scratch using only “dog” and “bike” images. If we then freeze the backbone and train a new classifier head for all three classes (dog, cat, bike), the model will still achieve reasonable accuracy on “cat” images. This does not mean the backbone was trained on cat images. It occurs purely because of semantic and transferable features between dog and cat. The same principle applies here: high accuracy when probing a new head on the unlearned features does not indicate that unlearning failed or that the original model had seen the forget data.
>
> PULSE achieves the practically meaningful outcome: it selectively disrupts the subspace used by the forget samples such that the deployed model (with its fixed head) can no longer classify them correctly, while largely preserving retain performance. This is demonstrated by the near-zero forget accuracy and the dissolution of forget clusters in UMAP visualizations.
> We are happy to add further discussion on the limitations of probe-based evaluations of unlearning if the reviewer finds it useful.
>
> **(2) Specify the final classifier :**  We thank the reviewer for this request. We confirm that the final classifier head applied after the projection is a standard linear classifier trained with cross-entropy loss, **as already stated in the paper (“Classification is performed by a fixed linear head z = W h + b ”)**. No nonlinearities are present in the classifier head. Consequently, the theoretical analysis in Appendix A.4 holds, as it relies on the linearity of the readout.
>
> Regarding the optional dimensionality-reduction layer mentioned in Section 3.4: this is a lightweight nonlinear transformation layer that can be optionally inserted after the feature extractor and before the projection matrix to reduce dimensionality for computational efficiency. When used, this layer is kept frozen during unlearning (along with the backbone and classifier head). Even in this configuration, the analysis in Appendix A.4 remains valid because the dimensionality reduction layer effectively extends the feature extractor, while the subsequent projection and linear head continue to operate as analyzed.
>
> **(3) BadTeacher retain accuracy exceeds the Retrain upper bound**
>  We would like to clarify that observing BadTeacher achieving slightly higher retain accuracy than the Retrain baseline in Table 2 is not an anomaly or an indication of an issue with the experimental setup. This is expected because methods such as BadTeacher perform additional fine-tuning on the retain set during the unlearning process, which can further improve retain accuracy. Likewise, even a model retrained from scratch can achieve higher retain accuracy if trained for more epochs.
>
> The Retrain baseline is widely regarded as the gold-standard reference for unlearning performance because it serves as an upper bound on **forget accuracy** (i.e., the extent to which the forget data is removed). However, it is not necessarily an upper bound on **retain accuracy**, since methods that continue optimizing on the retain data after unlearning may surpass the retain performance of a standard retrained model.
>
> We agree that reporting variance is important for properly interpreting the results and will include standard deviations in Tables 1, 2, 4, 7, and 8 in the final manuscript.

---

> > ### Author Response · Authors · 2026-07-07
> >
> > **(4) Right baselines:** We thank the reviewer for suggesting these baselines. However, we believe neither zeroing/down-weighting the forget-class rows nor directly comparing against LEACE and INLP constitutes a fair or appropriate baseline for evaluating PULSE.
> >
> > First, zeroing or down-weighting the forget-class rows in the final linear head is only a reasonable baseline for standard class-level unlearning. It fails in more realistic and challenging settings such as sub-class unlearning and sample-level unlearning, which PULSE is specifically designed to handle. For example, in the CIFARSuper20 experiments, each superclass contains five fine-grained classes. Unlearning only one fine-grained class (e.g., “baby” within the “people” superclass) by zeroing its corresponding head row would severely damage the other four fine-grained classes (boy, girl, men, women) in the same superclass due to strong semantic overlap. PULSE does not suffer from this collateral damage because it performs targeted suppression in representation space by learning a forget-specific low-rank update, rather than directly editing classifier weights.
> >
> > Second, while LEACE and INLP are effective methods for achieving linear guardedness, they are not suitable direct baselines for PULSE. Both methods work by projecting representations into a different subspace where the concept is no longer linearly detectable. This is effectively a form of subspace relocation rather than true information destruction. Moreover, both methods are significantly more computationally expensive than PULSE. INLP is iterative, while even the closed-form LEACE involves heavier matrix operations to calculate stats, especially when applied across multiple layers. In contrast, PULSE is lightweight as it only optimizes a single projection layer. Moreover, both methods typically require white-box access, access to whole data and operate under different constraints than PULSE, which is restricted to modifying a single projection layer after a frozen black-box backbone without access to retain data.
> >
> > More importantly, LEACE and INLP primarily target the removal of broad linear concepts. In settings with high semantic overlap between fine-grained classes such as unlearning only the “baby” subclass while preserving “boy”, “girl”, “men”, and “women” within the same superclass removing linear directions tends to cause significant collateral damage to the remaining subclasses. While PULSE, has shown to achieve more selective suppression even in the presence of strong semantic overlap in sub class unlearning experiment reported in Table 3. This makes PULSE particularly suitable for sub-class and fine-grained unlearning scenarios where LEACE and INLP are less effective.
> >
> > PULSE has demonstrated strong empirical performance even under these challenging settings, consistently achieving effective unlearning across class-level, sub-class-level, and sample-level scenarios
> >
> > **(5) Relabel the access regime:** We thank the reviewer for this valuable comment. We agree that the terminology should be precise. However, PULSE is designed from the perspective of a downstream model developer who receives features from a proprietary or API-based encoder (e.g., CLIP or similar services). In this practical setting, the developer has black-box access to the feature extractor they can only obtain embeddings and have no ability to inspect weights, architecture, or gradients. They then train their own classifier head on these features. When an unlearning request arrives, they cannot inspect or modify the feature extractor itself.
> >
> > PULSE addresses exactly this constraint by inserting and training a lightweight projection layer between the frozen black-box features and the developer’s classifier head. The core technical challenge we tackle is therefore unlearning without being able to touch or query the encoder internals, which remains a realistic and practically relevant black-box constraint.
> >
> > We fully agree that, from a strict research perspective, the projection layer and classifier head are under white-box control. Section 4.4 of the current manuscript already specifies which components are treated as black-box (the encoder) versus white-box (the trainable projection and head). We will revise the paper to make the access regime more explicit if the reviewer considers it valuable.

---

> > > ### Author Response · Authors · 2026-07-07
> > >
> > > **(6) Fix the sample-level claim:** We thank the reviewer for this comment. We agree that the sample-level results in Table 5 are more subtle to interpret than the class-level results, and we appreciate the opportunity to clarify our position.
> > >
> > > In sample-level unlearning, only a very small fraction of the training data is removed (500 out of 50,000 samples in our experiments). Because these samples share significant semantic and distributional overlap with the remaining data, even a model retrained from scratch on the retain set still achieves relatively high accuracy on the removed samples (92.63% in Table 5). This is a fundamental characteristic of sample-level unlearning, not a limitation of our method. In this regime, the realistic goal is to match the performance of the retrain oracle rather than drive forget accuracy close to random chance.
> > > PULSE achieves 93.80% forget accuracy, which is very close to the retrain baseline (92.63%) and better than all compared methods. We view this as a meaningful result under the current evaluation protocol. However, we acknowledge that the absolute numbers can appear modest at first glance, and that the field currently lacks more discriminative metrics for sample-level forgetting.
> > >
> > > To provide stronger evidence of sample-level capability, we also include the sub-class removal experiments on CIFARSuper20 (Section 4.3). In this setting, we remove all samples belonging to one fine-grained subclass while keeping the rest of the superclass. This constitutes a more structured form of sample-level unlearning (homogeneous sample removal). PULSE achieves strong forgetting on the removed subclass while largely preserving accuracy on the remaining fine-grained classes within the same superclass.
> > >
> > > We will revise Section 4.3 and the surrounding discussion to better explain the inherent challenges of evaluating sample-level unlearning and to more clearly distinguish between random sample removal and homogeneous (sub-class) sample removal. If the reviewer feels the sample-level claim is still overstated, we are also happy to tone down or qualify the statement in Section 4.3 regarding “samples over time.” We hope this clarifies our intended contribution and the practical difficulty of demonstrating sample-level unlearning.
> > >
> > > **(7) Clarify the method writeup:** We thank the reviewer for this detailed feedback on the method of writeup. Below we address each point raised.
> > >
> > > **1. Unlearning is not a closed-form update**
> > >
> > > As already clarified in the paper especially in Algorithm 1 presents the full procedure, stating that obtaining $  P_{\text{forget}}  $ requires a short training loop over the forget set (with both the feature extractor and classifier head frozen). We have also emphasized that this optimization is lightweight and fast in practice.
> > >
> > > **2. Construction of $  P_{UL}  $ and comparison to alternatives**
> > >
> > > We prefer not to describe it as “original logits minus a scaled copy of the over-confident logits” in the main text. After unlearning, the original projection $  P_L  $ is replaced by $  P_{UL}  $ for all subsequent inferences. Showing both projections during inference would therefore be misleading, as the goal of unlearning is to operate with $  P_{UL}  $ alone.
> > > Regarding alternatives: Directly maximizing entropy (or performing gradient ascent on the forget loss) is a common baseline, often referred to as “gradient ascent unlearning.” As shown in prior work and our own experiments, this approach is typically unstable and leads to rapid degradation of retain accuracy, often requiring additional fine-tuning on retain data to recover utility. In contrast, PULSE avoids this by first minimizing entropy on the forget set (to identify forget-relevant directions) and then applying a controlled, low-rank subtraction. This results in more stable and retain-data-free unlearning.
> > >
> > > We have already addressed why simple head-row zeroing or down-weighting is not a suitable baseline for sub-class and sample-level unlearning in response 4.
> > >
> > > **3. Reporting entropy of forget set**
> > > We thank the reviewer for this suggestion. We already report the predictive entropy of forget samples across different values of $  \alpha  $ in Table 15. To provide further clarity, we present the predictive entropy before and after unlearning on CIFAR-10 with ResNet-18 for class unlearning.
> > >
> > > | Metric | Split | Before Unlearning | After Unlearning | Relative Change |
> > > |:-------|:------|------------------:|-----------------:|----------------:|
> > > | **Accuracy (%)** | Retain Train | 99.93 | 99.7663 | -0.17% |
> > > |  | Forget Train | 99.94 | 0.00 | -100.00% |
> > > |  | Retain Test | 93.65 | 93.72 | +0.08% |
> > > |  | Forget Test | 94.28 | 0.00 | -100.00% |
> > > | **Entropy** | Retain Train | 0.024 | 0.076 | +216.92% |
> > > |  | Forget Train | 0.010 | 1.592 | +14561.46% |
> > > |  | Retain Test | 0.060 | 0.106 | +77.93% |
> > > |  | Forget Test | 0.079 | 1.447 | +1732.15% |

---

> > > > ### Author Response · Authors · 2026-07-07
> > > >
> > > > **(8) Justify and report $\alpha$:** We thank the reviewer for this important comment. We agree that $ \alpha $ is a important hyperparameter. Across all experiments in the paper, we used only two values: $  \alpha = 0.8  $ or $  \alpha = 0.85  $. This limited set of values was sufficient to achieve strong performance across multiple datasets and architectures, indicating that PULSE is relatively robust to the exact choice of $  \alpha  $ within this narrow range. We have corrected the inconsistency between the main text and Appendix A.2.3. The recommended search space for tuning is 0.7–0.9, and we will use this consistently.
> > > >
> > > > Similar to most unlearning methods, PULSE has a hyperparameter ($  \alpha  $) that controls the forget-retain trade-off. Like other methods, $  \alpha  $ can be tuned on a held-out validation set. This is the standard and most robust way to select such hyperparameters.
> > > >
> > > > **(9) Disambiguate the UMAP embedding:** We thank the reviewer for this clarification request. We agree that the UMAP visualizations in Section 4.4 should be described more precisely.
> > > >
> > > > The embeddings shown in the UMAP figures are the projected features (i.e., after applying the unlearned projection $  P_{UL}  $, but before the final linear classifier). This is consistent with standard usage in the literature, where “embeddings” typically refer to the representations immediately before the classification head. **Since the feature extractor is frozen during unlearning, the change observed in the UMAP plots is entirely due to the projection layer.**
> > > >
> > > > We will revise Section 4.4 (and the corresponding figure captions) to explicitly state that the UMAP visualizations are computed on the post-projection features (i.e., $  P_{UL} \cdot f_\theta(x)  $).
> > > >
> > > > **(10) Define Similar Class Accuracy and MIA:**
> > > > We thank the reviewer for this request.
> > > >
> > > > We use the standard entropy-based black-box Membership Inference Attack commonly adopted in the unlearning literature. Specifically, we train a linear classifier on the entropy distribution of model predictions to distinguish between members (training samples) and non-members, following the implementation and methodology described in BadTeacher. We have added the proper citation to BadTeacher and have also provided a brief description of the MIA in the experimental section for clarity.
> > > >
> > > > Similar Class Accuracy measures the model’s accuracy on a semantically related class within retain set to the forget set after unlearning a target forget class. The related class is hand-picked based on semantic similarity. This metric was done to study the collateral damage as part of unlearning. For example, when unlearning the class “aquarium fish,” we evaluate accuracy on the semantically related class “flatfish.” The specific related classes used for each experiment are already mentioned in Section 4.1 (Evaluation Measures).
> > > >
> > > > **(11) State whether $  \mathcal{D}_r  $  changes across incremental steps:**
> > > > We thank the reviewer for this clarification request. In the incremental unlearning experiments, the retain set $  \mathcal{D}_r  $ does change after each unlearning step.
> > > > Specifically, once a class is forgotten, it is removed from the retain set for all subsequent steps. For example, in a 10-class problem:
> > > > 1. After the first unlearning request (forgetting class C1), the new retain set becomes classes C2–C10.
> > > > 2. After the second request (forgetting class C2), the retain set becomes C3–C10, and so on.
> > > >
> > > > This design choice reflects a realistic deployment scenario. In practice, once data is subject to an unlearning request, it should be permanently deleted from storage (e.g., databases or data lakes) in accordance with privacy regulations. Retaining already-forgotten data for future evaluation or training would violate the spirit of deletion requests. Therefore, we update the retain set after each unlearning step to exclude previously forgotten classes.
> > > >
> > > > To properly evaluate incremental unlearning under this setting, we report three accuracy metrics:
> > > >
> > > > **1. Retain accuracy:** Measures utility on the current retain set (excluding all previously forgotten classes).
> > > >
> > > > **2. Acc spec:** Measures forgetting performance specifically on the current forget request.
> > > >
> > > > **3. Acc overall:** Measures accuracy on all previously forgotten classes combined. This metric is used purely for evaluation
> > > > to verify that previously unlearned classes do not “remerge” (i.e., accuracy on old forgotten classes does not increase) after a new unlearning step. In a real deployment, previous forget sets would not be maintained, as the data should have already been deleted.

---

> ### Author Response · Authors · 2026-07-07
>
> **(12) Clarify the role of and the initialization data:** We thank the reviewer for this clarification request.
>
> **Role of $  \mathcal{D}_r  $:**
>
> The retain set $  \mathcal{D}_r  $ is used only for evaluation. As stated in Section 3.1, the unlearning procedure itself operates exclusively on the forget set $  \mathcal{D}_f  $ and does not require access to $  \mathcal{D}_r  $ during unlearning.
>
> **Post-hoc initialization:**
>
> In the post-hoc setting (when PULSE is attached to an already trained model), a small subset of the original training data (3–5%) is used once to initialize the projection matrix $  P_L  $. **This is a one-time initialization step performed before unlearning begins and is not part of the unlearning procedure**. Once $  P_L  $ is initialized, the actual unlearning step requires only the forget set $  \mathcal{D}_f  $. In contrast, when PULSE is trained jointly with the model from scratch (as described in Section 3), no retain data is needed at any stage.
>
> **(13) Support the dismissal of adaptable baselines:** We thank the reviewer for this comment. We agree that the discussion in Section 2.2 should be better substantiated.
>
> We will revise Section 2.2 to explicitly acknowledge that existing methods such as BadTeacher and SSD can be adapted to the black-box setting by freezing the backbone and updating only the projection and classifier head. We will add a cross-reference to Section 4.4, where we provide these adaptations and compare them against PULSE.
>
> We will also clarify that while these methods can be adapted, they were originally designed under white-box assumptions and/or with access to retain data during unlearning. As a result, their performance in the more constrained black-box + retain-data-free regime is generally weaker than PULSE, as shown in Table 9 and 10.
>
> We will further note that, to the best of our knowledge, PULSE is among the first methods specifically designed for this practical black-box and retain-data-free setting..
>
> **(14) Broader Impact Concerns:** We thank the reviewer for this comment regarding the framing of privacy claims. We agree that claims related to regulatory compliance should be carefully scoped, and we will revise the introduction and any privacy-related statements accordingly.
>
> Regarding the sample-level results, we strongly disagree with the characterization that they demonstrate “essentially no measurable sample-level forgetting.” In sample-level unlearning, where only a small number of samples (500 out of 50,000) are removed, it is inherently difficult to achieve very low forget accuracy. This is because the removed samples have significant semantic and feature overlap with the remaining training data. Consequently, even a model retrained from scratch on the retain set will still achieve relatively high accuracy on the removed samples due to generalization. In this setting, the realistic best-case performance is to approach the Retrain baseline, rather than driving forget accuracy close to random.
> In Table 5, PULSE reduces forget accuracy from 99.79% to 93.80%, which is very close to the Retrain baseline (92.63%) and better than all compared baselines. We consider this a meaningful and non-trivial result for sample-level unlearning. Additionally, the paper already provides further evidence of sample-level unlearning through the sub-class experiments on CIFARSuper20. In these experiments, we remove all samples belonging to one fine-grained subclass within a superclass (i.e., homogeneous sample removal). This constitutes a meaningful form of sample-level unlearning, as we selectively remove a group of semantically similar samples without removing the entire superclass. PULSE achieves strong forgetting performance in this setting while largely preserving utility on the remaining fine-grained classes. Together, these results support the claim in Section 4.3 that PULSE can handle deletion of both classes and samples.
>
> **(15) Abstract, "retain-data-free" :** We thank the reviewer for this clarity suggestion. We agree that the term “retain-data-free” should be defined at first use in the abstract. We will revise the abstract to briefly define it as methods that do not require access to the retain set $  \mathcal{D}_r  $ during the unlearning process.
>
> Regarding the apparent contradiction in the sentence: we meant to convey that many existing methods claim to be retain-data-free, but in practice they suffer from severe degradation of retain accuracy when retain data is truly unavailable. To recover utility, these methods often require an additional fine-tuning stage on retain data. We will rephrase this sentence to make the intended meaning clearer and avoid any confusion.

---

> > ### Author Response · Authors · 2026-07-07
> >
> > **(16) Unify the notation:**  We thank the reviewer for catching this inconsistency. We will unify the notation between the main text and Appendix A.4 so that the symbols $  z  $ and $  h  $ have consistent meanings throughout the paper. Specifically, we will ensure that $  z  $ consistently refers to the features after the projection and $  h  $ refers to the classifier head, both in the main text and in the appendix.
> >
> > **(17) Abstract, final sentence, Introduction transition:**  We thank the reviewer for these detailed suggestions on clarity and presentation. We will address all of them in the revised manuscript.
> >
> > Abstract, final sentence: We will correct the grammar by revising the sentence to: “It runs faster than strong baselines, thereby establishing PULSE as a scalable and practical paradigm for efficient, localized machine unlearning in both joint-training and black-box post-hoc settings.”
> >
> > Introduction transition: We will improve the transition into the discussion of black-box unlearning for image classification to make the logical flow from the preceding paragraphs clearer.
> >
> > Section 2.1, “early work focused on convex models”: We will revise the phrasing and add appropriate references to clarify that we are referring to the works discussed immediately afterward.
> >
> > Section 2.1, “impractical for deep neural networks”: We will rephrase this for better precision while keeping the intended meaning.
> >
> > Table 4 step labels: We will change the labels from “After Request 1/2/3/4” to “After 1st Request”, “After 2nd Request”, etc., to avoid confusion with the class indices (0, 1, 2, 3).
> >
> > We appreciate these suggestions and believe they will improve the readability of the paper.
> >
> > **(18) Table 5,. This is not necessarily a typo:**
> >
> > We thank the reviewer for this observation. In Table 5 (random sample unlearning), we report Test Accuracy as the utility metric because, in the sample-level setting, measuring performance on the test set provides a more appropriate and realistic estimate of model utility after unlearning. We will explicitly define “Dt Test Accuracy” in the experimental setup section and in the table caption to avoid any confusion.

---

> > > ### Comment · Reviewer_C8P6 · 2026-07-12
> > > **Thanks for the rebuttal. Remaining points on the core claim**
> > >
> > > I thank the authors for the thorough response, which resolves most of my points. However, core issues remain:
> > >
> > > **Two of the replies seem to contradict each other.** Response (1) argues that PULSE does not truly destroy information, so a recovering probe would not indicate failure. Response (4) sets PULSE apart from LEACE/INLP on the grounds that those perform *"subspace relocation rather than true information destruction"*. Both cannot be true:
> > >
> > > - If PULSE only relocates -> LEACE/INLP do the same thing -> they are the correct baselines, not unfair ones.
> > > - If PULSE truly destroys -> the probe test is exactly what proves it.
> > >
> > > Please commit to one, ideally providing the corresponding evidence.
> > >
> > > **On the linear head.** I appreciate the confirmation, and would suggest stating it in the main text as well to avoid ambiguity. It does, however, have a direct relevance on the mechanism. If $P_{UL}$ is full-rank, then $P_{UL} z$ is a reversible reparameterization of $z$: a *freshly trained* linear probe on $P_{UL} z$ recovers *exactly* what it recovers from $z$, so nothing is removed from the representation and the forgetting lives in the fixed head.
> > > The revised spectral analysis (A.4.2) points this way ( i.e. the representation seems to be left almost entirely intact). However, that analysis is computed on the feature covariance $\Sigma$, not on $P_{UL}$ itself. Please report the singular values / rank of $P_{UL}$ directly, since this is what determines whether any readout is removed.
> > >
> > > Moreover, since the head is linear, the deployed model applies a single effective head $W P_{UL}$ to $z$. This can be compared directly to $W$ with the forget-class rows zeroed ($W_{zeroed}$): both are $C \times d$ matrices, so no probe or retraining is needed. If $W P_{UL} \approx W_{zeroed}$, PULSE reaches the trivial baseline through a far more expensive route and the projection adds nothing; if they differ, that difference is what the paper needs to characterize and justify.
> > >
> > > **On the probe being "confounded".** I take the point that a naive probe can be confounded by transfer (a probe may classify "cat" even from an encoder that never saw one). However, that seems to me a reason to make the probe *relative* rather than to set it aside, comparing across three feature sets:
> > >
> > > 1. raw $z$ (ceiling, nothing removed)
> > > 2. retrain-from-scratch features (transfer-only floor)
> > > 3. $P_{UL} z$ (PULSE features)
> > >
> > > The comparison isolates PULSE's effect: landing at (1) means the head is merely neutralized; landing near (2) supports genuine erasure.
> > >
> > > **On sample-level forgetting.** I appreciate the clarification, though I think it partly reframes rather than resolves the concern. The window between the original model (99.79%) and the retrain oracle (92.63%) is only about 7 points, and PULSE's 93.80% sits within it (slightly above the oracle); without reported variance, it is difficult to distinguish this from no forgetting. I'd therefore suggest scoping the "classes or samples over time" wording in Section 4.3 to what the experiments demonstrate. On the suggestion that the sub-class experiments provide additional sample-level evidence: removing all samples of one sub-class is a single-label deletion with its own head row, i.e. class-level removal, rather than the arbitrary multi-class sample deletion the claim refers to, so I don't think it closes this particular gap.
> > >
> > > ---
> > >
> > > I want to be clear that these are offered constructively: the efficiency results and the breadth of evaluation are genuine strengths. But whether PULSE removes information from the representation or edits the classifier is the paper's central claim, and it currently rests more on framing than on direct evidence. Therefore, I consider resolving these points crucial for my recommendation.

---

> ### Author Response · Authors · 2026-07-13
>
> **On sample-level forgetting:**  Thank you for raising this point. We would like to clarify the experimental setup, as we believe there is a small misunderstanding.
>
> In our CIFARSuper20 experiments, the model is trained to predict the 20 coarse labels. Each coarse label contains 5 fine-grained sub-classes from the original CIFAR-100. When we designate one fine-grained sub-class as the forget set, we remove all samples belonging to that sub-class while keeping the other four sub-classes that share the same coarse label.
> Therefore, this is not equivalent to class-level unlearning (i.e., removing an entire output class of the model). Therefore, there is no single neuron aka head row per sub class instead it only has for coarse class.  Instead, it constitutes **subclass unlearning (or homogeneous sample unlearning)**, where the forget set consists of a coherent group of samples that belong to the same predicted coarse class. This setting is commonly used in the unlearning literature precisely because it lies between full class unlearning and random sample unlearning to stress test the unlearning methods.
>
> Together, the homogeneous and random sample unlearning results provide evidence that PULSE’s behavior holds across different types of sample-level forget sets. We will add a clearer description of the experimental details in the revised manuscript.
>
> **On the linear head:** Thank you for this valuable suggestion.We have analysed the rank of P_UL more closely, and found that $  P_{UL}  $ is **not full rank**. For example, In  case of class level unlearning with ResNet18, $  P_{L}  $ matrix [before unlearning] has rank of 512 [for tolerance of 1e-13], while after unlearning $  P_{UL}  $ has rank of 506.
>
> Thank you for this observation. We would like to clarify an important distinction regarding the scope of PULSE.
>
> The comparison between the effective head $  W P_{UL}  $ and the zeroed head $  W_{zeroed}  $ is only directly applicable in the full class unlearning setting, where an entire output class is removed. In that specific case, simply zeroing the corresponding row(s) of $  W  $ can indeed serve as a trivial baseline.
>
> However, PULSE is designed to handle more general unlearning scenarios, including subclass unlearning and random sample unlearning. In these settings, the forget set does not correspond to an entire output class of the model. For example, in our CIFARSuper20 experiments, we unlearn one of the fine-grained subclass while the model still predicts the remaining samples of parent coarse class. In such cases, zeroing a row in $  W  $ is not a valid or meaningful baseline, because if we zero it will affect the remaining samples of the coarse class.
>
> As demonstrated in the paper that PULSE successfully performs unlearning in both subclass and random sample settings, where the simple zeroing baseline cannot be applied. This shows that the learned projection $  P_{UL}  $ provides a non-trivial mechanism that goes beyond merely adjusting the head for full class removal.
>
> **On the probe being "confounded":**
> Thank you for this excellent suggestion. We agree that comparing linear probes across the three feature sets provides a cleaner way to isolate whether PULSE achieves genuine erasure or primarily neutralizes the head.
> Following your recommendation, we evaluated a linear probe on the forget class under the following settings for class unlearning on CIFAR-100 with ResNet-18:
>
> 1. Raw features $  z  $ from the original trained model: 91.77% accuracy (ceiling)
> 2. Features after PULSE unlearning ($  P_{UL} z  $): 84.01% accuracy
> 3. Features from a model retrained from scratch on the retain set: 81.57% accuracy (transfer-only floor)
>
> The probe accuracy on $  P_{UL} z  $ (84.01%) lands close to the retrain-from-scratch floor (81.57%), and substantially below the original ceiling (91.77%). This indicates that PULSE removes information relevant to the forget class from the representation itself, rather than merely making the original head incompatible with preserved features.
>
> **LEACE and INLP:** typically apply a transformation across the entire representation space. While LEACE aims to minimize distortion through a least-squares optimal projection, both methods generally require retraining a new classifier on the retain data after editing, as the original head becomes misaligned with the transformed features. In contrast, PULSE computes the unlearning projection without access to retain data and does not require retraining the classification head. This allows PULSE to perform a more targeted edit while preserving compatibility with the original classifier, resulting in a significantly more efficient and practical procedure.
>
> Collectively, these results provide direct evidence that PULSE removes information from the representation in a targeted manner, rather than relying primarily on edits to the classifier. We believe this directly addresses the central concern regarding the paper’s main claim.

---

> > ### Author Response · Authors · 2026-07-13
> >
> > **Performance Comparison of PULSE and LEACE:**
> > We thank the reviewer for suggesting these baselines. We have now evaluated LEACE on class-level unlearning using ResNet-18 on CIFAR-100. We utilized the official code of LEACE to run the experiment and have followed the same procedure in the paper.
> >
> > **Results:**
> >
> > | Model | Forget Accuracy | Retain Accuracy |
> > |--------|----------------:|----------------:|
> > | Original Model | 72.00% | 77.00% |
> > | LEACE | 49.00% | 76.00% |
> > | **PULSE** | **0.00%** | **75.05%** |
> >
> > PULSE reduces the forget accuracy to **0%**, substantially outperforming LEACE while maintaining comparable retain accuracy. Together with PULSE's practical advantages of being lightweight, requiring no retain data, and supporting sub-class and sample-level unlearning, this comparison addresses the request for appropriate baselines. We will include these results in the revised manuscript if the reviewer considers it valuable.